# Synthetic lethality between TP53 and ENDOD1

Zizhi Tang [1,6], Ming Zeng [1,6], Xiaojun Wang [1,6], Chang Guo[1], Peng Yue[1], Xiaohu Zhang[1], Huiqiang Lou[2], Jun Chen [3], Dezhi Mu[1], Daochun Kong[4], Antony M. Carr [5✉] & Cong Liu[1,5✉]

The atypical nuclease ENDOD1 functions with cGAS-STING in innate immunity. Here we identify a previously uncharacterized ENDOD1 function in DNA repair. ENDOD1 is enriched in the nucleus following $H_2O_2$ treatment and $ENDOD1^{-/-}$ cells show increased PARP chromatin-association. Loss of ENDOD1 function is synthetic lethal with homologous recombination defects, with affected cells accumulating DNA double strand breaks. Remarkably, we also uncover an additional synthetic lethality between ENDOD1 and p53. ENDOD1 depletion in $TP53$ mutated tumour cells, or p53 depletion in $ENDOD1^{-/-}$ cells, results in rapid single stranded DNA accumulation and cell death. Because $TP53$ is mutated in ~50% of tumours, ENDOD1 has potential as a wide-spectrum target for synthetic lethal treatments. To support this we demonstrate that systemic knockdown of mouse $EndoD1$ is well tolerated and whole-animal siRNA against human $ENDOD1$ restrains $TP53$ mutated tumour progression in xenograft models. These data identify ENDOD1 as a potential cancer-specific target for SL drug discovery.

[1] Department of Paediatrics, SCU-CUHK Joint Laboratory for Reproductive Medicine, Key Laboratory of Birth Defects and Related Diseases of Women and Children (Ministry of Education), West China Second University Hospital, Sichuan University, 610041 Chengdu, China. [2] School of Life Sciences, China Agricultural University, 100193 Beijing, China. [3] College of Life Sciences, Zhejiang University, 310058 Hangzhou, China. [4] School of Life Sciences, Peking University, 100871 Beijing, China. [5] Genome Damage and Stability Centre, School of Life Science, University of Sussex, Falmer BN1 9RQ, UK. [6]These authors contributed equally: Zizhi Tang, Ming Zeng, Xiaojun Wang. ✉email: a.m.carr@sussex.ac.uk; congliu@scu.edu.cn

The development of PARP inhibitors (PARPi), such as Olaparib[1] and Talazoparib[2], to treat BRCA-deficient breast cancer[3,4] opened up a new therapeutic strategy for cancer subtype-specific chemotherapy, synthetic lethality[5–7]. As our understanding of PARP and its inhibition has developed, the range of cancers considered for PARPi therapy has expanded to include homologous recombination deficient (HRD) tumours. Significant interest in potential new synthetic lethal (SL) targets has led to many drug development programs aiming to identify molecules that inhibit proteins found to be SL with specific genetic backgrounds common to defined tumour subtypes. An ideal SL target would be mutated or silenced in a wide spectrum of tumours.

During a characterization of PARPi-induced DNA damage responses in the presence of hepatitis B virus oncoprotein HBx expression, which renders cells HRD by sequestering Cullin4-DDB1 and thus depleting CRL4$^{WDR70}$ [8], we initially identified an increase in endonuclease domain-containing protein 1 (ENDOD1) peptides following six days of Olaparib treatment (Supplementary Fig. 1a). This prompted us to characterize a potential role in DNA repair for ENDOD1 in more detail.

## Results

**ENDOD1 functions in DNA single-strand break repair.** ENDOD1 has previously been identified as a 500 amino acid protein that interacts with RNF26 to modulate the cGAS-STING innate immunity pathway[9]. ENDOD1 contains three C-terminal transmembrane motifs, a single endonuclease domain (residues 49–257) and an N-terminal signal peptide (residues 1-22) (Fig. 1a). Upon immunoblotting, both endogenous and ectopically expressed ENDOD1 displayed multiple bands (Supplementary Fig. 1b–c). This indicates the presence of multiple isoforms, potentially derived from post-translational processing and/or modification. In addition, upon expression of either C- or N-terminal Flag-tagged constructs, only the C-terminal, but not the N-terminal, construct could be detected with α-Flag. This suggests that the signal peptide is cleaved as predicted (Supplementary Fig. 1c). In untreated RPE1 cells, indirect immunofluorescent staining for ENDOD1 was predominantly cytoplasmic, but following high-dose hydrogen peroxide (H$_2$O$_2$) treatment, we identified the formation of α-ENDOD1 reactive foci in nuclei that are absent in ENDOD1 knockout RPE1 cells (ENDOD1$^{-/-}$), (Fig. 1b). This implies that certain forms of ENDOD1 can enter the nucleus and access damaged DNA.

While ENDOD1 was first identified as a cytoplasmic protein[9], a recent mass spectrometry study identified ENDOD1 peptides in the nucleus[10]. In the context of our identification of damage-induced intra-nuclear ENDOD1 foci, this is consistent with an additional role for ENDOD1 in DNA repair. Indeed, ENDOD1$^{-/-}$ cells showed moderate sensitivities to CPT, cisplatin and the G4 inhibitor Cx5461. No detectable sensitivity was observed upon IR or HU treatment (Supplementary Fig. 1d). Unexpectedly, ENDOD1$^{-/-}$ cells displayed obvious resistance to the single-strand break (SSB)-inducing agent H$_2$O$_2$. A similar H$_2$O$_2$ resistance was observed for GES-1, a normal gastric epithelial line, following siRNA against ENDOD1.

The kinetics of ENDOD1 foci induced by H$_2$O$_2$ treatment resembled that of poly(ADP-ribose) (PAR) foci, whose signals appeared prior to those of ENDOD1 (Fig. 1b). The focal signals of ENDOD1, which clearly overlap with PAR-positive foci (Supplementary Fig. 2a), required PARP activity: siRNA knockdown of PARP1/2, or treatment with PARPi, diminished the H$_2$O$_2$-induced ENDOD1 foci (Supplementary Fig. 2b). No effect was seen for siRNA knockdown of PARP3. Using single cell electrophoresis under denaturing conditions (alkaline comet assay) to assess the

repair kinetics of gaps and nicks, we did not detect a major repair defect between proliferating ENDOD1$^{-/-}$ and RPE1 control cells (Fig. 1c). However, when cells were arrested in G1 phase by serum starvation before treatment, a modest but statistically significant defect was seen in repair kinetics. Consistent with this, despite the clearance of PAR foci in ENDOD1$^{-/-}$ being higher in the early phase of repair, PAR foci loss over time was slower in ENDOD1$^{-/-}$ serum starved (G1 arrested) cells when compared with control RPE1 cells. We also noted a modest increase of PAR foci in unperturbed serum starved (G1 arrested) ENDOD1$^{-/-}$ cells (Supplementary Fig. 2c, control untreated lanes). This may represent a low-level of PARP-responsive DNA lesions accumulating in G1 phase that are eventually eliminated before or when cells progress into S/G2. Taken together, these data are consistent with ENDOD1 influencing PARP-dependent SSB repair.

**ENDOD1 influences PARP chromatin association.** Interestingly, PARP1 knockdown with two different siRNAs significantly reversed the increased H$_2$O$_2$ resistance of ENDOD1$^{-/-}$ cells (Fig. 1d), whereas knockdown of either PARP2 or PARP3 (which plays a minor role in SSB repair[11]) did not affect the resistance (Supplementary Fig. 2d). We therefore asked if PARPi could similarly reverse the relative H$_2$O$_2$ resistance of ENDOD1$^{-/-}$ cells. Despite the fact that treating the parental RPE1 cells with a combination of PAPRi and H$_2$O$_2$ was toxic (Supplementary Fig. 2e), the H$_2$O$_2$ resistance observed in ENDOD1$^{-/-}$ cells was not reversed. These distinct outcomes of siPARP1 and PARPi treatment implies that the protection against oxidative stress caused by ENDOD1 deletion is attributable to the physical presence of PARP1 per se, rather than its enzymatic activity. Indeed, in ENDOD1$^{-/-}$ cells 30 min after acute treatment with methyl methanesulfonate (MMS) we observed that PARP1 and PARP3 were enriched[12] in the insoluble histone-containing nuclear fraction (Fig. 1e). In contrast to ENDOD1$^{-/-}$ cells, control RPE1 cells retained only residual PARP in this fraction. Taken together, we conclude ENDOD1 contributes to prevent excessive PARP association with damaged DNA. It is unclear how loss of ENDOD1 manifests as increased resistance to oxidative stress.

Previous work has identified HR factors, including BRCA1, BRCA2 and Fanconi Anaemia proteins, as being required for DNA repair following PARP inhibition and that compromising the HR pathway results in synthetic lethality with PARPi[3]. To examine how ENDOD1 interplays with homologous recombination factors and PARP we assayed cell survival following siRNA of either WDR70, BRCA1, BRCA2, ARID1A, BLM, CTIP, CHK1, EXO1, MRE11 or FANCA in either ENDOD1$^{-/-}$ or control RPE1 cells (Fig. 2a). The profile of SL upon depletion of HR factors in ENDOD1$^{-/-}$ cells mirrored that for PARP inhibition. Co-depleting ENDOD1 and BRCA1 in RPE1 cells using three independent ENDOD1 siRNA's showed similar synthetic lethality (Supplementary Fig. 3a). In contrast, a range of HR-competent non-cancer cells (including RPE1 and GES-1) exhibited no proliferation defects upon siRNA ablation of ENDOD1 (Fig. 2b).

PARPi-induced HRD cell death is known to coincide with the generation of toxic DNA structures[13]. Consistent with this, BRCA1 siRNA treatment of ENDOD1$^{-/-}$ cells resulted in increased γH2AX and 53BP1 foci, common markers of DSBs (Fig. 2c and Supplementary Fig. 3b), and elevated chromosomal aberrations (Fig. 2d), mimicking the consequences of PARPi treatment of HRD cells[13]. Similarly, depleting BRCA2 in ENDOD1$^{-/-}$ cells elevated 53BP1 foci formation (Fig. 2c and Supplementary Fig. 3c). ENDOD1 - HRD SL effects were dependent on the presence of PARPs (Fig. 2c and Supplementary Fig. 3c) but were not reversed by PARPi treatment

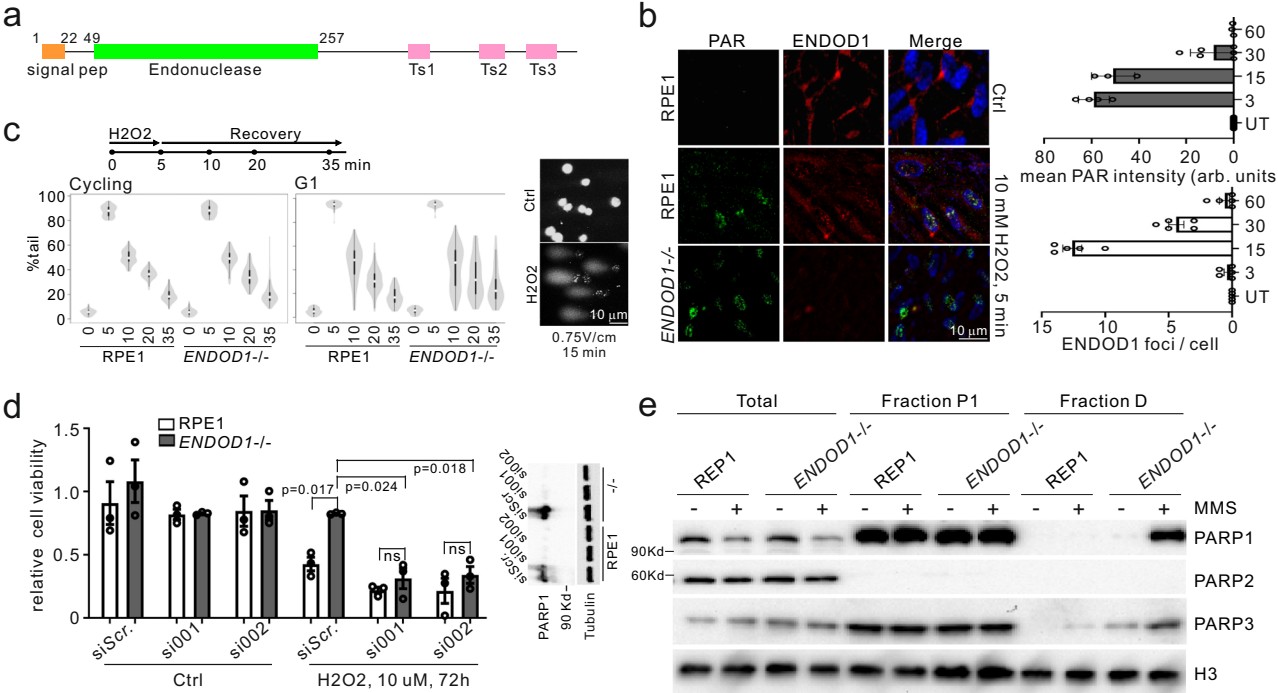

**Fig. 1 Characterization of ENDOD1 in DNA repair. a** Schematic of ENDOD1 protein showing the predicted signal peptide (residues 1–22), the endonuclease domain (residues 49–257) and three C-terminal transmembrane motifs (Ts). **b** Left: Representative indirect immunofluorescence images for α-ENDOD1 (ABclonal) and α-PAR: untreated RPE1 cells (top row), RPE1 and $ENDOD1^{-/-}$ cells after a 5 min treatment with 10 mM H2O2 (bottom two rows). For the merged panels nuclear DNA was counterstained with DAPI. Right: quantification for nuclear signals at the indicated time points after H2O2 treatment. arb. units: arbitrary units. $n = 3$–5 biologically independent experiments. **c** Comet assay to assess repair efficiency. Left: quantification of tail moments from three repeats for proliferating RPE1 and $ENDOD1^{-/-}$ cells at the indicated time points after H2O2 challenge ($p = 0.197$ for RPE 35v $ENDOD1^{-/-}$ 35, Kruskal test). Middle: tail moments from 150 cells from each of 3 biologically independent experiments for serum-starved G1 arrested RPE1 and $ENDOD1^{-/-}$ cells at indicated time points after H2O2 challenge ($p = 0.787$ for RPE1 20 v $ENDOD1^{-/-}$ 20; $p = 0.0000159$ for RPE1 35v $ENDOD1^{-/-}$ 35, Kruskal test). Right: representative images of alkaline comet assay. Percentage tail moment was calculated by dividing the pixel intensity of tails by that of heads. White dot: median. Thick whisker: third quartile. Thin whisker: upper/lower adjacent values (1.5x inter-quartile range). **d** Relative viability of RPE1 or $ENDOD1^{-/-}$ cells treated with two different siPARP1 (si001 and si002) 48 h before challenge, with or without continuous H2O2 treatment (10 µM). Assay: CCK8 colorimetry. Inset showing the knockdown efficiency of each siRNA. $n = 3$ biologically independent experiments. Significance test: two-tailed Student's $t$ test. $p$ values: 0.0057, 0.047. ns: not significant. **e** Whole-cell extract (total), nuclear extracts (P1) and MNase-digested extract (fraction D) from RPE1 or $ENDOD1^{-/-}$ cells probed for PARP1, PARP2 and PARP3. Cells were treated with 0.01% MMS for 30 min. Histone H3 serves as a control. Representative image of 3 independent experiments.

(Supplementary Fig. 3d), indicating that they are a result of "PARP trapping" and not PARP activity[13]. The cytotoxicity and DSB generation upon concomitant inhibition of HR and ENDOD1 could be reproduced in HRD breast cancer cell line MCF-7 (ref. [14]) (Supplementary Fig. 3e, f). Thus, we conclude that ablation of ENDOD1 phenocopies PARPi in compromising the genomic integrity of HR-defective cells.

**ENDOD1 is SL with TP53.** We next tested a panel of cell lines to establish the SL profile of siENDOD1 treatment in cancer cells (Fig. 3a, Supplementary Fig. 4a and Supplementary Table 1). Unexpectedly, in addition to preventing proliferation of HRD cancer cells, siENDOD1 also inhibited the proliferation of multiple non-HRD cancer cells. Upon further analysis this SL correlated with TP53 status (Fig. 3b), a potentially important observation. The TP53 mutations spanned the common mutation hotspots (i.e. R248, R273 and R280)[15]. Cell lines, including A549 and MDA-MB-361 that do not carry TP53 or HR mutations were not sensitive to siENDOD1. To rule out off-target effects we demonstrated that siENDOD1-induced toxicity in the C33A cancer cells (TP53-R273C) could be reproduced with distinct siRNAs (Supplementary Fig. 4b). We also validated the SL between p53 and ENDOD1 using our $ENDOD1^{-/-}$ and RPE1 control cells. While siRNA control treated $ENDOD1^{-/-}$

and siTP53 treated RPE1 cells were viable, siTP53 treated $ENDOD1^{-/-}$ cells arrested in G1 within 60 h and cell death became apparent at 5 days and was extensive after 7 days (Fig. 3c). Concomitant treatment of RPE1 cells with siENDOD1 and siTP53 also revealed SL (Supplementary Fig. 4c). The cell death correlated with nuclear abnormalities and markers of apoptosis (Supplementary Fig. 4d). Knockdown of TP53 exacerbated the G1 cell cycle arrest that is already apparent in $ENDOD1^{-/-}$ cells (Supplementary Fig. 4e). Importantly, the toxicity of siTP53 to $ENDOD1^{-/-}$ cells can be rescued by ectopic expression of ENDOD1 full-length protein (Supplementary Fig. 4f).

The killing effect of siTP53 treatment of $ENDOD1^{-/-}$ cells was also apparent in G1-arrested non-cycling cells (Fig. 3d) that do not incorporate BrdU and lack bulk DNA synthesis (Supplementary Fig. 4g), suggesting that the cytotoxicity is independent of DNA replication. The SL between ENDOD1 and TP53 was also recapitulated in coisogenic HCT116 cells (colon cancer): HCT116 cells expressing wildtype p53 remained viable, but those expressing an isoform of p53 (Δ40p53)[16] did not proliferate (Supplementary Fig. 4h). Taken together, these data indicate ENDOD1 ablation, in addition to killing cells with HR defects due to chromatin-associated PARP, is synthetic lethal to cells with functional loss of TP53 even in the absence of DNA replication.

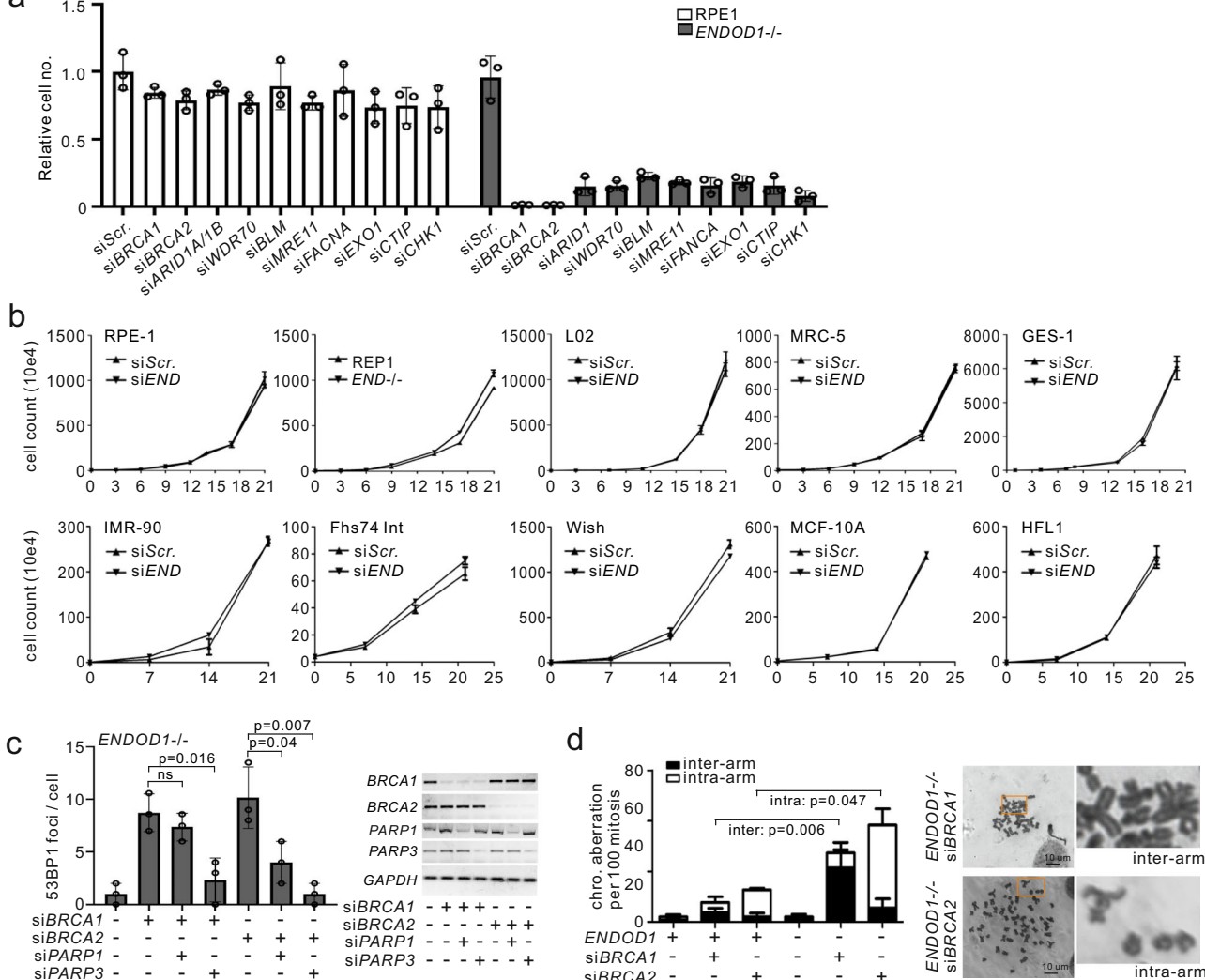

**Fig. 2 Depletion of *ENDOD1* causes SL with HRD. a** Relative viability of *ENDOD1*$^{-/-}$ and control RPE1 cells subjected to the indicated siRNA treatments. siRNA was transfected three times with three-day intervals. Cell viability was determined by CCK8 colorimetric assay. $n = 3$ biologically independent experiments. **b** Proliferation was quantified by haemocytometer cell counting over 3 weeks for the indicated immortalized non-cancerous cells with control siRNA (si*Scr.*) or si*ENDOD1* treatment every 3 days. Control experiments using RPE1 and *ENDOD1*$^{-/-}$ are shown in the first two panels. **c** Left: quantification of 53BP1 foci in *ENDOD1*$^{-/-}$ cells 72 h following treatment with the indicated siRNAs. $n = 3$ biologically independent experiments. Error bars: SEM. Significance test: two-tailed Student's *t* test. ns: not significant. Right: semi-quantitative PCR showing knockdown efficiency. **d**. Ablation of *BRCA1* or *BRCA2* in *ENDOD1* null cells results in increased chromosome aberrations. Left: quantification of inter- and intra-chromosomal aberrations in RPE1 or *ENDOD1*$^{-/-}$ cells treated with si*BRCA1* or si*BRCA2*. $n = 3$ biologically independent experiments. Error bars: SEM. Significance test: two-tailed Student's *t* test. Right; representative images.

***ENDOD1 TP53* SL correlates with ssDNA formation**. To characterize the SL interaction between ENDOD1 and p53 we used comet assays to asses DNA lesions occurring in G1 arrested *ENDOD1*$^{-/-}$ and control RPE1 cells following si*TP53*. Alkaline comet assays revealed that si*TP53* induced a high level of DNA breaks in *ENDOD1*$^{-/-}$ but not RPE1 cells (Fig. 4a and Supplementary Fig. 5a). Neutral comet assays did not show an increase in DNA damage in the same experiment. Thus, these breaks were likely in the form of SSBs. Consistent with this, analysis of phosphorylated RPA32 Serine 33 (pRPA32), a typical marker of single-stranded DNA (ssDNA), showed that pRPA32 staining was elevated in the nuclei of either proliferating or G1 arrested *ENDOD1*$^{-/-}$ si*TP53* cells, but was not elevated in the RPE1 cells treated with si*TP53* or *ENDOD1*$^{-/-}$ cells co-transfected with control siRNA (Fig. 4b and Supplementary Fig. 5b). Non-denaturing α-BrdU staining showed evidence of ssDNA tracts in

si*TP53* treated *ENDOD1*$^{-/-}$ cells, but not control RPE1 cells (Fig. 4c) and the α-BrdU signal was sensitive to S1 nuclease treatment (Fig. 4d). The production of non-denatured α-BrdU staining upon si*TP53* treatment of *ENDOD1*$^{-/-}$ cells was reproduced with a second si*TP53*, and could be eliminated by re-introducing *ENDOD1* upon lentiviral infection (Supplementary Fig. 5c, d). Consistent with the production of ssDNA, S1 nuclease preferentially digested genomic DNA extracted from si*TP53* treated *ENDOD1*$^{-/-}$ cells when compared to relevant controls (Fig. 4e and Supplementary Fig. 5e). In contrast to these indicators of ssDNA lesions, DSB surrogate markers, γH2AX and 53BP1 foci, were not significantly increased in G1-arrested *ENDOD1*$^{-/-}$ cells 48–96 h after si*TP53* transfection (Supplementary Fig. 5f, g). We conclude that concomitant loss of ENDOD1 and p53 functions results in the generation of ssDNA and cell death.

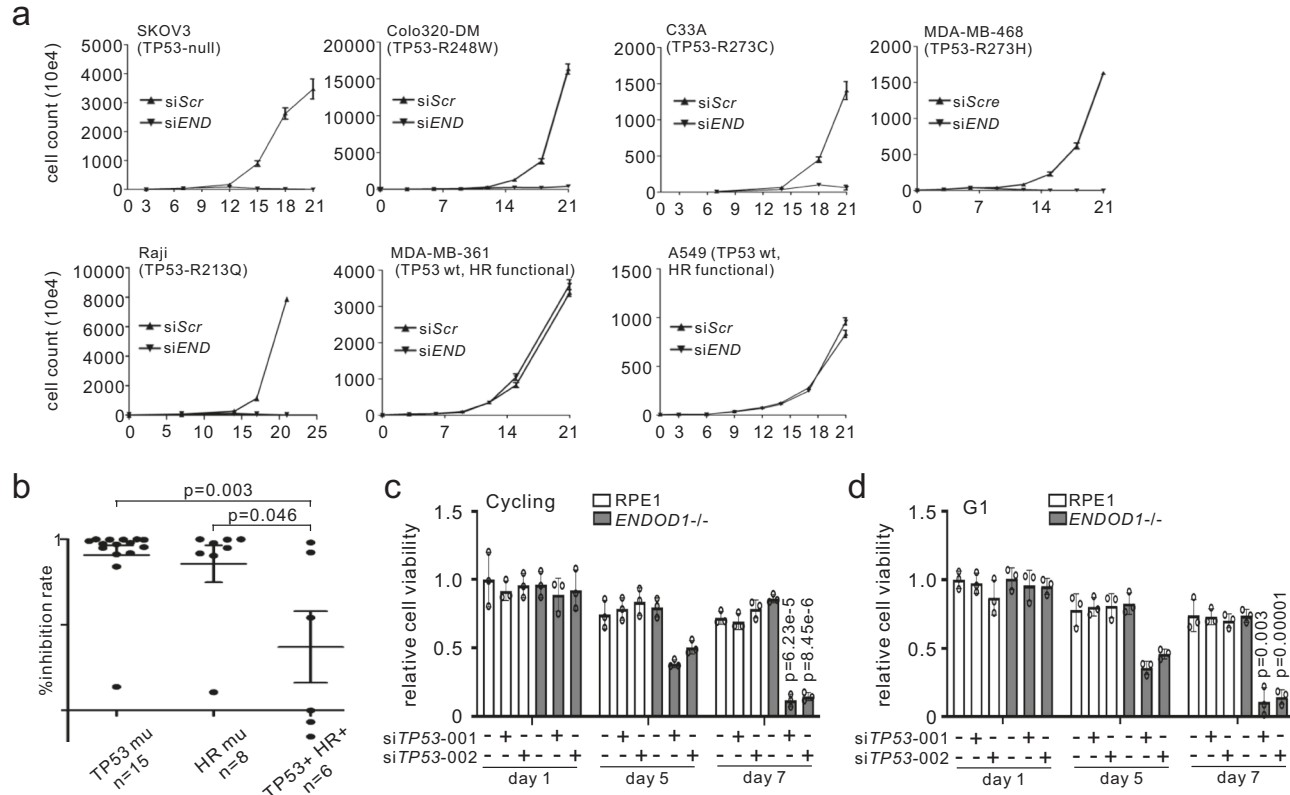

**Fig. 3 Concomitant loss of ENDOD1 and p53 causes SL. a** Proliferation curves for representative cancer cell lines following either siENDOD1 or control transfection. $n = 3$ biologically independent experiments. Error bars: SEM. At 3-day intervals cells were passaged, counted by haemocytometer and transfected. Reported TP53 status is indicated. **b**. Correlation of TP53/HR mutations with cytotoxicity of siENDOD1 in terms of inhibition rate (%) for cancer cell lines (each point represents mean of $n = 3$ biologically independent experiments). Error bars; SEM. **c** Relative viability determined by CCK8 colorimetry of ENDOD1$^{-/-}$ and control RPE1 cells 5 and 7 days after treatment with two different siTP53 (si001 and si002). $n = 3$ biologically independent experiments. Error bars: SEM **d** Equivalent experiment as in (**c**), using cells arrested in G1 by serum starvation. $n = 3$ biologically independent experiments. Error bars: SEM. All significance tests: two-tailed Student's $t$ test.

**TP53 hotspot mutations permit ssDNA production upon ENDOD1 ablation.** TP53 mutations in human cancers exhibit diverse functional consequences including loss- and gain-of-function[17] and almost invariably display abnormal DNA binding and transcriptional properties[18]. A recent study reported rapid PARP-dependent recruitment of p53 to sites of DNA damage that required the DNA binding and carboxyl terminal domains of p53[19]. We therefore assayed the production of pRPA32 foci upon siENDOD1 in a range of cell lines with defined TP53 mutations. siENDOD1-induced pRPA32 foci were increased in serum-starved (G1 phase) cancer lines harbouring a TP53 null allele (SKOV-3 and NCI-H1299) in addition to cancer cells harbouring gain-of-function (GOF) mutants R273C (C33A) and R273H (MBA-MD-468). No increase in signal was apparent when the TP53 competent line A549 was treated with siENDOD1 (Fig. 4f).

To establish if this phenomenon in a broader range of TP53 mutations we complemented a TP53 null cell line, SKOV-3, with either wild-type or domain-specific and hotspot mutants and tested for pRPA32 foci upon siENDOD1 treatment (Fig. 4g and Supplementary Fig. 5h). Wild type TP53 prevented ssDNA formation as expected. However TP53 mutants affecting the DNA binding domains (R175H, L194F, R248Q, R248W, R273H, R273C, R280K), the oligomerization domain (L344P), the putative nuclease activity (H115N)[20] or the non-specific nucleic acid binding activity (C-terminal deletion 363–393)[21], could not prevent ssDNA production. A transactivation domain mutation (LW22/23QS) showed an intermediate phenotype. Notably, the gain-of-function (GOF) mutants (i.e. R273C, R273H and R280K)

in the DNA binding domain[22] were, like the loss-of-function DNA binding domain mutants, not capable of preventing pRPA32 foci formation. This indicates that both decreased and increased p53 DNA binding causes cytotoxicity upon ENDOD1 ablation. Collectively, these data show that ENDOD1 inhibition is toxic to cells bearing TP53 mutations that span the common hotspot sites.

**PARylation is required for ssDNA production.** We next characterized the requirements for the generation of ssDNA. Dual treatment of ENDOD1$^{-/-}$ cells with siTP53 and either siPARP1/2 or siPARP3 significantly reduced pRPA32 foci formation compared to controls (Fig. 5a). Ablation of either PARP1/2 or PARP3 function suppressed pRPA32 foci when the TP53 mutated cancer cell lines C33A, MBA-MD-468, SKOV-3 and NCI-H1299 were treated with siENDOD1 (Supplementary Fig. 6a-b). We subsequently examined the chromatin association of PARP1 when p53 was depleted in ENDOD1$^{-/-}$ cells. siTP53 treatment of RPE1 control cells resulted in a modest increase in chromatin-associated PARP1. Untreated ENDOD1$^{-/-}$ cells already displayed a modest level of chromatin associated PARP1. Treatment of ENDOD1$^{-/-}$ cells with siTP53 generated a pronounced increase in the chromatin association of both PARP1 and PARP3 (Fig. 5b), suggesting that ssDNA formation and cell inviability are dependent on PARP association with chromatin.

Quantifying PAR staining after siTP53 treatment of RPE1 cells showed that the ablation of p53 alone did not significantly affect PARylation, but siTP53 treatment of ENDOD1$^{-/-}$ cells resulted

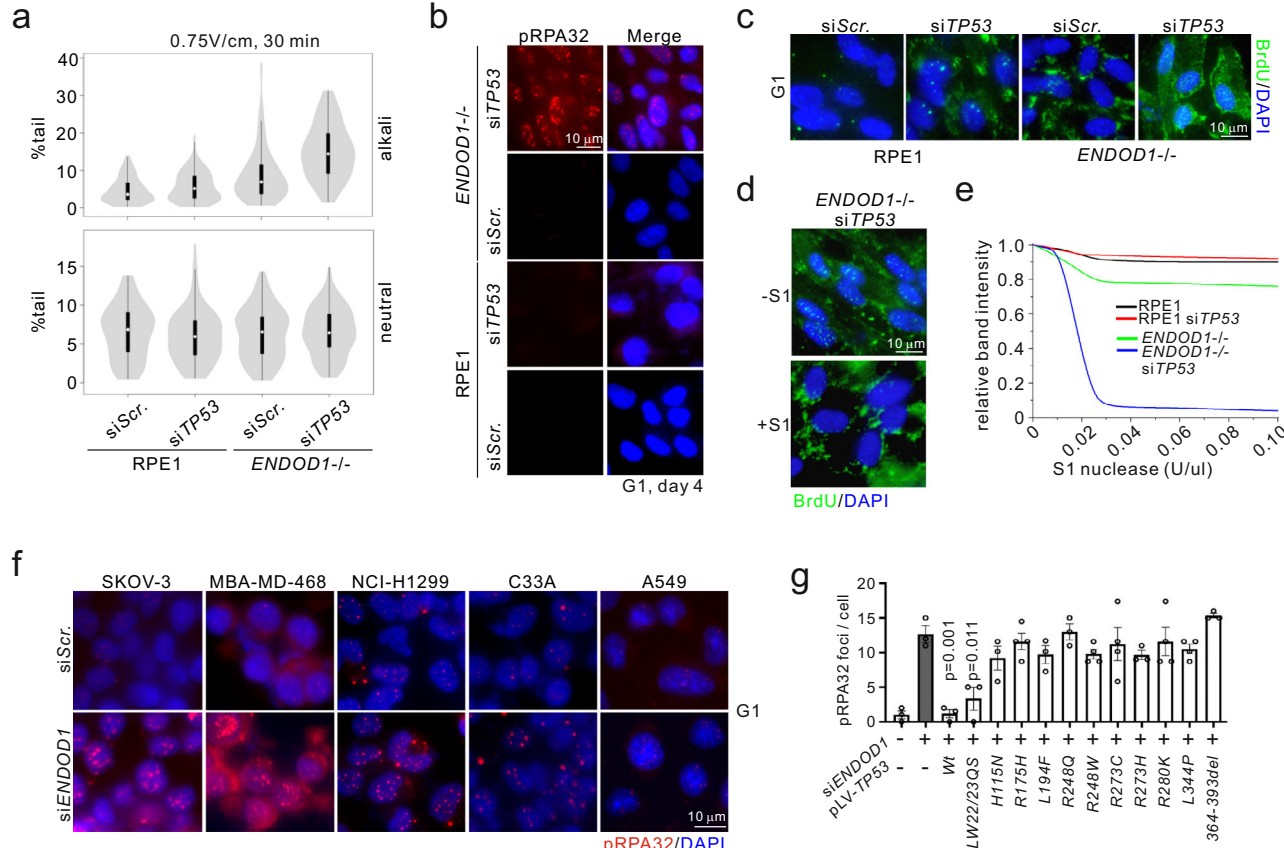

**Fig. 4 SL between *ENDOD1* and *TP53* correlates with ssDNA formation. a** Quantification of tail moments for neutral (bottom) and alkali (top) comet assays in G1-arrested *ENDOD1⁻/⁻* and control RPE1 cells treated with si*TP53* or control siRNA. n = 150 cells from each of 3 biologically independent experiments. White dot: median. Thick whisker: third quartile. Thin whisker: upper/lower adjacent values (1.5x inter-quartile range). **b** Immunofluorescent staining for pRPA32 in *ENDOD1⁻/⁻* and control RPE1 G1 arrested cells 96 h following si*TP53* or control treatment. Merged image is with DAPI staining. **c** Equivalent experiment as in (**b**), non-denatured staining for BrdU that was incorporated into cells before serum starvation. **d** Non-denatured BrdU signals in *ENDOD1⁻/⁻* si*TP53* cells with or without prior S1 nuclease digestion. Representative of 3 independent experiments. **e** Quantification of agarose gel band intensity of undigested genomic DNA purified from the indicated cells after treatment with the specified units of S1 nuclease. Representative image of 3 independent experiments. A representative original gel for 0.02 U/μl is shown in Supplementary Fig. 5e. **f** pRPA32 staining in cancer cell lines that harbour *TP53* mutations 72 h after treatment with control (si*Scr.*) or si*ENDOD1*. A549 is a *TP53* wild type control. **g**. Quantification of pRPA32 foci in si*ENDOD1* treated SKOV-3 cells complemented with the indicated *TP53* alleles. n = 3–4 biologically independent experiments. Error bars; SEM. Significance test: two-tailed Student's t test.

in a significant increase in nuclear PAR in both cycling and G1 arrested cells (Fig. 5c). Co-staining for pRPA32 and PAR showed that pRPA32 signals overlapped with PAR (Fig. 5d). This indicates that chromatin-associated PARP1 is active upon the concomitant loss of ENDOD1 and p53. Importantly, the specificity to ENDOD1 and p53 of these phenomena were validated in SKOV-3 (*TP53* null) cells: the emergence of pRPA32 and PAR foci upon si*ENDOD1* was significantly reduced following expression of siRNA-resistant *ENDOD1* (Supplementary Fig. 6c) and, like pRPA32 foci formation (cf. Fig. 4g), PAR foci were significantly reduced by the expression of wild type *TP53* (Fig. 5e and Supplementary Fig. 6d). Consequently, toxicity of si*ENDOD1* to SKOV-3 was reverted by *ENDOD1* or *TP53* expression (Supplementary Fig. 6e). With the exception of *TP53-LW22/23QS* (which also significantly reduced pRPA32 foci formation), mutant alleles of *TP53* did not prevent PAR foci following si*ENDOD1* (Fig. 5e). These data imply that p53 supresses PARylation and that, like the effect on pRPA32, this is independent of transactivation.

The DSBs induced by PARPi treatment of HRD cells are not dependent on PARP activity[13]. Consistent with this, as we showed above, PARPi treatment did not reduce the 53BP1 foci

observed in *ENDOD1⁻/⁻* si*BRCA1* cells (cf. Supplementary Fig. 3d). Intriguingly, and in contrast to this, inhibition of PARP activity with either Talazoparib or Olaparib prevented the formation of pRPA32 foci in G1-arrested cells (Fig. 5f) and significantly reduced cell killing upon si*TP53* treatment of *ENDOD1⁻/⁻* cells (Fig. 5g). The same is observed in multiple *TP53*-deficient cancer lines treated with si*ENDOD1* (Supplementary Fig. 6f). When *ENDOD1⁻/⁻* cells were treated with Talazoparib for the first 72 h after si*TP53* treatment and then the Talazoparib was removed (time 0) the nuclear PAR signal became evident at 6 h and was highly induced at 8 and 12 h (Fig. 5h). pRPA32 foci lagged behind, but reached the levels seen in cells not treated with Talazoparib by 12 h. The Inhibition of parylation by either si*PARP1* or PARPi also strongly inhibited ssDNA formation, as evidenced by the reduction of the non-denatured BrdU signal (Supplementary Fig. 6g). Interestingly, PARPi only attenuated, but did not eliminate, the S1-sensitivity of DNA isolated from si*TP53* treated *ENDOD1⁻/⁻* cells (Supplementary Fig. 6h). This may reflect that DNA lesions (i.e. SSBs) are present, but not processed into longer gaps, when cells are treated with PARPi. This suggests that long-tract ssDNA is only one form of DNA damage in si*TP53* treated *ENDOD1⁻/⁻* cells. Thus,

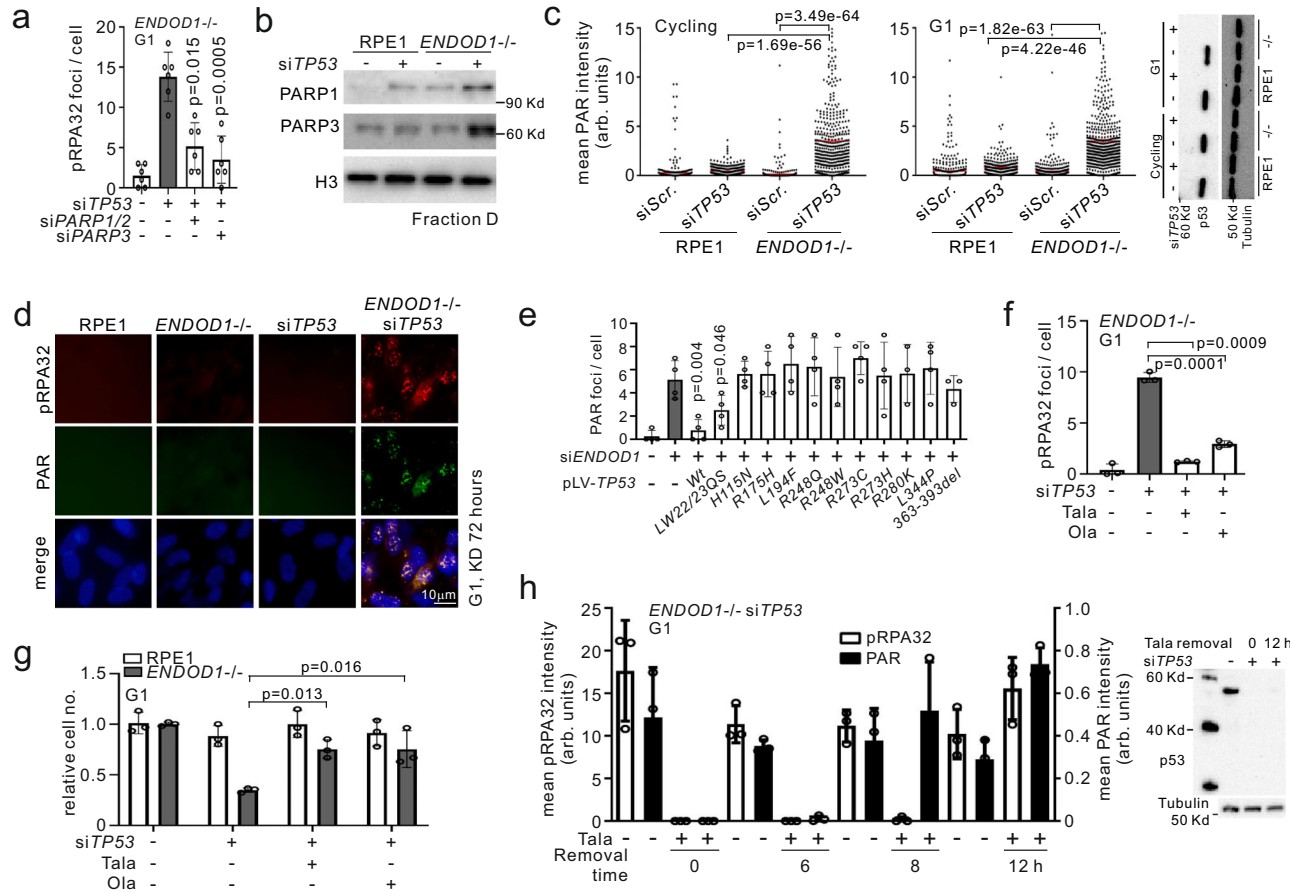

**Fig. 5 PARP activity is required for the SL between *ENDOD1* and *TP53*. a** pRPA32 foci in serum starved *ENDOD1*$^{-/-}$ cells treated with the specified siRNAs. $n = 6$ biologically independent experiments. Error bars: SEM. **b**. Immunoblotting for PARP1 on chromatin (fraction D—see Fig. 1e) in *ENDOD1*$^{-/-}$ and control RPE1 cells following si*TP53* or siRNA control treatment. Representative image of 4 independent experiments. **c** Quantification of nuclear PAR staining 72 h following the indicated siRNA treatments of *ENDOD1*$^{-/-}$ and control RPE1 cells (left: cycling, middle: G1 arrested, right: immunoblot showing knockdown efficiency for si*TP53*). arb. units: arbitrary units. $n = 381$–731 cells. Red bar: median. Whiskers: SEM. **d** pRPA32 and PAR co-staining 72 h following the indicated siRNA treatments of *ENDOD1*$^{-/-}$ and control RPE1 cells. Representative image of 3 independent experiments. **e** Quantification of PAR foci in SKOV-3 cells upon exogenous expression of the indicated *TP53* alleles. arb. units: arbitrary units. $n = 3$–4 biologically independent experiments. **f** pRPA32 foci 3 days after serum starved *ENDOD1*$^{-/-}$ cells were treated with si*TP53* and PARPi (10 nM Talazoparib, 100 nM Olaparib). $n = 3$ biologically independent experiments. Error bars: SEM. **g** Relative proliferation 6 days after *ENDOD1*$^{-/-}$ and control RPE1 cells were treated with si*TP53* and PARPi (2 nM Talazoparib, 20 nM Olaparib) $n = 3$ biologically independent experiments. Error bars: SEM. **h** Talazoparib addition-removal assay in *ENDOD1*$^{-/-}$ cells. 10 nM Talazoparib was added upon si*TP53* transfection and removed 72 h later (time 0). Cells were fixed at the indicated time for α-pRPA32 or α-PAR staining. Immunoblot evaluation for the knockdown efficiency of *TP53* is shown on the right. arb. units: arbitrary units. $n = 3$ biologically independent experiments. Error bars; SEM. All significance tests: two-tailed Student's $t$ test.

unlike the SL observed between PARPi—HRD and *ENDOD1*—HRD, the SL between *ENDOD1*—*TP53* requires PARP activity as well as its physical presence.

**XRCC1 is required for PARP activation and ssDNA formation.** Using the same Talazoparib withdrawal protocol we next examined the requirement for single-strand break repair in ssDNA generation in G1 arrested cells. Upon withdrawal of Talazoparib, concomitant knockdown of si*TP53* with *XRCC1*, the scaffold of the SSB machinery, eliminated pRPA32 foci in *ENDOD1*$^{-/-}$ cells. si*TDP1* showed a modest reduction in foci and si*LIG3* did not have an impact (Fig. 6a). Nucleotide excision repair (NER) factors (XPA and XPC) did not influence pRPA32/PAR (Supplementary Fig. 7a). Consistent with this we observed that si*XRCC1* suppressed the formation of PAR foci upon si*TP53* in *ENDOD1*$^{-/-}$ cells (Fig. 6b and Supplementary Fig. 7b). Importantly, Talazoparib treatment did not prevent the formation of XRCC1 foci in si*TP53* treated *ENDOD1*$^{-/-}$ cells (Fig. 6c). These results indicate that

XRCC1 stimulates PAR catalysation by chromatin-associated PARP only when ENDOD1 and p53 are simultaneously absent. It also suggests that XRCC1-mediated break repair triggers ssDNA production.

**ssDNA production requires resection factors.** To determine if the mechanism of ssDNA generation by PARP chromatin association in G1 phase involves the canonical DNA end resection enzymes that usually process DSBs in S/G2 phase of the cell cycle, we examined key resection factors in G1-arrested si*TP53* treated *ENDOD1*$^{-/-}$ cells subjected to PARPi treatment and withdrawal. Knockdown of a range of resection factors, including MRE11, NBS, CTIP, BLM, EXO1, BRCA1 and FANCA suppressed pRPA32 foci formation, but not PAR formation, when G1 arrested *ENDOD1*$^{-/-}$ cells were treated with si*TP53* (Fig. 6d). Analysis for foci formation of resection factors identified an accumulation of MRE11, CTIP, BLM, FANCD2, NBS1 and BRCA1 foci (Supplementary Fig. 7c). Like the formation of PAR foci (cf. Fig. 5h), MRE11 foci formation

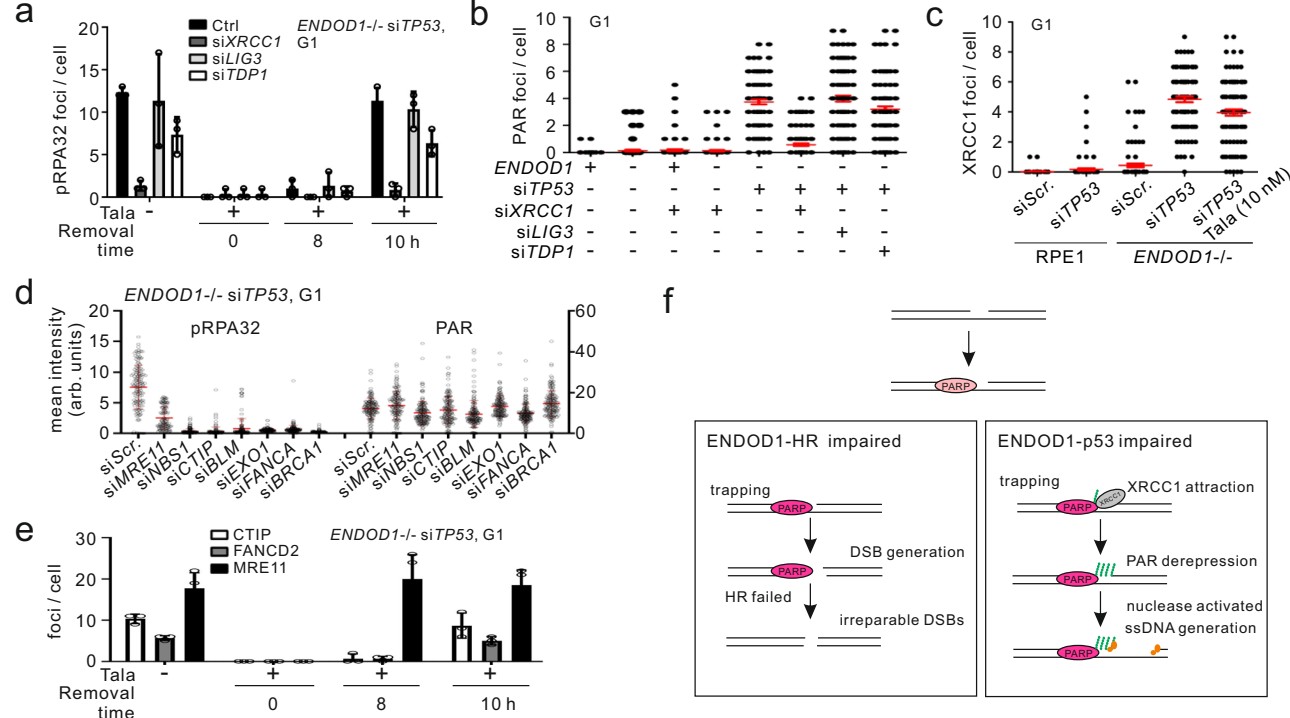

**Fig. 6 SSBR and HR machinery provides key signals for ssDNA production. a** Quantification of pRPA32 foci in a Talazoparib addition-removal assay with siTP53 treated ENDOD1[−/−] cells co-treated with the indicated SSBR siRNAs. 10 nM Talazoparib was added upon siRNA treatment and removed 72 h later (time 0). n = 3 biologically independent samples. Error bars: SEM. **b** Quantification of a-PAR foci in G1 serum starved ENDOD1[−/−] and control RPE1 cells following treatment with the indicated siRNAs. n = (99-103) × 5 cells (each data point represents the average foci number of 5 cell counts). Red bar: median. Whiskers: SEM. **c** Quantification of XRCC1 foci in serum-starved ENDOD1[−/−] and control RPE1 cells following siTP53 treatment with or without co-treatment with Talazoparib. n = (99–100) × 5 cells (each data point represents the average foci number of 5 cell counts). Red bar: median. Whiskers: SEM. **d** pRPA32 and a-PAR foci in siTP53 treated ENDOD1[−/−] cells co-treated with and the indicated siRNAs targeting DNA end processing factors. n = 151–167 cells. **e** Quantification of CTIP, FANCD2 and MRE11 foci in a Talazoparib addition-removal assay with siTP53 treated ENDOD1[−/−] cells. n = 3 biologically independent experiments. Error bars: SEM. 10 nM Talazoparib was added upon siRNA treatment and removed 72 h later (time 0). **f** Schematic model for how ENDOD1 protects genomic integrity together with HR (left) and p53 (right). See discussion for details.

in siTP53 treated ENDOD1[−/−] cells following Talazoparib removal was rapid, with foci visible at 8 h (Fig. 6e), whereas the formation of CTIP and FANCD2 foci only became apparent at 10 h. This suggests that MRE11 as a key nuclease for generating the ssDNA. Interestingly, unlike α-NBS1 and α-BRCA1 antibodies, α-phospho-NBS1 and α-phospho-BRCA1 did not reveal foci (Supplementary Fig. 7c), despite being robust markers of resected DSB ends in S/G2[23,24]. These data suggest that inappropriate activity of resection factors that would usually process DSBs in S/G2 phase are producing ssDNA tracts, even in G1 cells, when both ENDOD1 and p53 functions are impaired (Fig. 6f). However, ssDNA production in the absence of p53 and ENDOD1 is mechanistically distinct from canonical DSB resection.

The impact of XRCC1 is dominant over the resection machinery as ablation of XRCC1 significantly reduced levels of MRE11 staining (Supplementary Fig. 7d), indicating that XRCC1 acts upstream of the resection machinery. Combining the results in Fig. 6c, we propose that the ssDNA production in ENDOD1-TP53 double mutant occurs in a stepwise manner: SSB-bound XRCC1 activates PARP1 that subsequently attracts the resection machinery to initiate long-tract ssDNA processing. Inhibition of PARP1 suppresses productive ssDNA generation but leaves SSBs unrepaired (cf. Supplementary Fig. 6h).

**ENDOD1 is a potential drug target.** ENDOD1 is SL both with HRD and with TP53 mutation, suggesting a wide range of potential target tumours. A key issue with drugs that target

specific proteins is systemic tolerance. To address this, C57/B6 mice were injected twice weekly with siRNA for the murine homolog of ENDOD1 (simEndod1) or a disease causing control, simWdr70. Efficacy of whole animal gene silencing was assessed by semi-quantitative PCR to be ~75% (Supplementary Fig. 8a). After 60 days, simWdr70 treated mice lost weight and were euthanized. 90 days into simEndod1 treatment littermates retained normal weight (Fig. 7a). Histological examination revealed minimal pathological alterations in simEndod1 treated animals (Fig. 7b), whereas simWdr70 treated littermates displayed increased cell debris, fibrosis and disorganized tissue structures in the lung, intestine and liver. Carditis and event heart failure can also complicate cancer treatment with high-dose chemotherapeutic regimens[25]. Hematoxylin and eosin staining of heart tissues from simEndod1 treated animal showed no myoper-icarditis that manifested with lymphocyte infiltration (Fig. 7b) and echocardiography showed normal cardiac function without hypertrophy in simEndod1 treated animals, with no changes to anatomic dimension or systolic and diastolic function (Supplementary Fig. 8b).

A common side effect of antineoplastic drugs is myelosuppression and cytopenia. In the peripheral blood of simEndod1 mice, no apparent cytopenia was observed (Fig. 7c) and populations of T cells (CD3e[+]), NK (NK1.1[+]) cells, granulocyte (Gr-1[+]) and macrophage (F480[+]CD11b[+]) were conserved (Fig. 7d). The lack of peripheral myelosuppression in simEndod1 is supported by the preservation of hematopoietic stem cells (HSC) in bone marrow

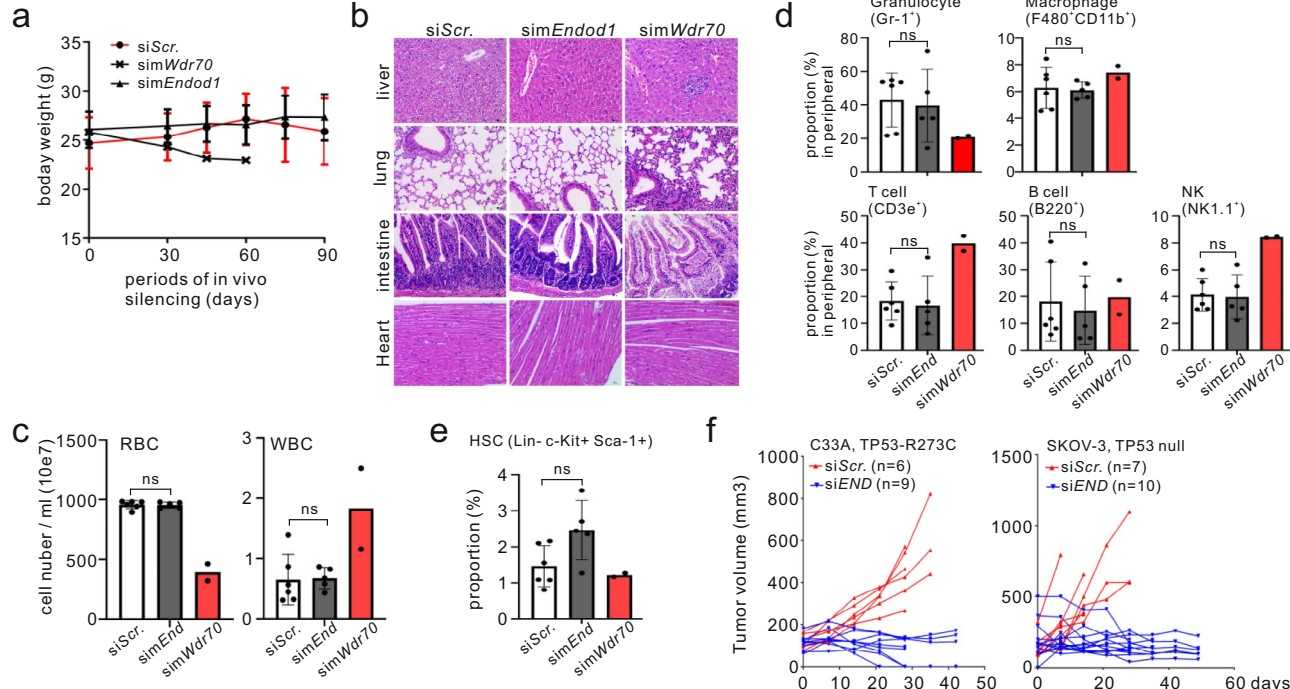

**Fig. 7 mEndod1 systemic knockdown is well tolerated. a** Body weight tracked through 90 days for the indicated in vivo knockdown groups (two injections per week). simWdr70 mice were sacrificed at 60 days due to severe disease. **b** Representative images of hematoxylin and eosin staining of paraffin-embedded sections from the indicated tissues. **c** Haemocytometer counts for peripheral blood cells at the endpoint of the experiment for each knockdown group. Control 90 days (n = 6 animals), simEndod1 90 days (n = 5 animals), simWdr70 60 days (n = 2 animals). Significance test: two-tailed Student's t test. ns: not significant. **d** FACS analysis for peripheral myeloid and lymphoid cells when experiments terminated. n = 6, 5 and 2 animals for simScr, simEnd and simWdr70, respectively. Error bars: SEM. Significance test: two-tailed Student's t test. ns: not significant. Cell surface markers used are shown in parentheses. **e** Equivalent FACS analysis as above for bone marrow HSC. **f** Anti-tumour treatment using whole animal in vivo knockdown of ENDOD1 for p53-deficient (SKOV-3 and C33A) xenograft models in nude mice. Volumes of individual tumours were measured. x-axis: treatment days. Numbers of animals (n) are indicated.

(Lineage$^-$ Sca1$^+$ c-Kit$^+$ cells: LSK) (Fig. 7e). Consistent with this, the multilineage potential of bone marrow was generally sustained for T/NK cells and myeloid compartments (granulocyte and macrophage) (Supplementary Fig. 8c). We conclude that the suppression of mEndod1 function does not cause short-term myelotoxicity (<3 months).

To establish the therapeutic potential of ENDOD1 xenograft tumour models were established from the *TP53* mutated SKOV-3 and C33A cancer cells and a *TP53* wild-type control cancer line, MDB-MA-361. Progression of SKOV-3 and C33A tumours was effectively curbed upon in vivo siENDOD1 treatment when compared to the siScr. control group (Fig. 7f). In contrast, siENDOD1 was ineffective in restraining disease progression of MDB-MA-361 Tumours (Supplementary Fig. 8d). Taken together, these data suggest that exploiting *TP53-ENDOD1* SL may yield therapeutic advantage, while minimizing unwanted side effects seen in conventional chemotherapy.

## Discussion
Previous reports identified ENDOD1 as an RNF26-interacting protein that modulates cGAS-STING-dependent innate immune signalling through an as yet uncharacterized mechanism[9]. Our identification of ENDOD1 during experiments that initially aimed to identify PARPi-dependent responses in a specific HRD cell line (HBx-induced CRL4$^{WDR70}$ defect) led us to characterize ENDOD1 in the context of DNA repair. We show that ENDOD1 loss results in increased PARP chromatin association and, as with PARPi treatment, this manifests in SL with HRD. This SL resembled that of PARPi treated HRD cells, involving the

accumulation of lethal DNA structures during replication that require resolution by HR-dependent pathways, and relying on the physical presence of PARPs rather than their activity (Fig. 6f, left). This suggests that targeting ENDOD1 in HRD cancers may provide an alternative to PARPi therapy.

Surprisingly, we identified a second distinct SL interaction with mut-p53 that encompasses the major cancer-specific hotspots. The incompatibility of *ENDOD1* ablation and *TP53* mutations was also linked to PARPs, but followed a distinct pattern: SL was evident in G1-arrested serum-starved cells, correlated with the formation of tracts of ssDNA and was dependent on both the physical presence of PARPs and their activity (Fig. 6f, right). Mechanistically we found that, in the absence of both ENDOD1 and p53, intrinsic PARP1 chromatin association was elevated and that XRCC1 was required to stimulate the catalytic activity of chromatin-associated PARPs in order to generate ssDNA and cell death. ssDNA generation was also dependent on the activity of the resection machinery that is usually inactive in G1. These observations define a PARP-dependent DNA damaging activity that is distinct from those engendered by PARPi. What structures are initiating PARP recruitment and activation, how this is prevented by ENDOD1 and p53, plus how the resection machinery is activated inappropriately remain important future questions.

*TP53* mutation occurs in the majority of human tumours and is associated with therapy-refractory malignancy. Previous efforts to exploit p53 in cancer treatment have largely focused on restoring the pro-apoptotic and cell cycle arrest potential of mutant versions[26]. Our work opens up the possibility that a wide range of *TP53* mutations, including common gain-of-function alleles, could be exploited using a SL approach to induce toxic

DNA lesions specifically in cancer cells, but not surrounding tissue. To establish if such an approach is feasible, we treated mice with sim*Endod1* to ascertain if the in vivo knockdown of m*En-dod1* could be tolerated. In contrast to whole animal ablation of m*Wdr70*, which acted as a disease-causing control, whole animal ablation of m*Endod1* was well tolerated, with minimal evidence of relevant tissue damage and acceptable levels of myelosuppression. We then established three xenograft models and demonstrated that whole animal treatment with human si*ENDOD1* resulted in profound disease control for two *TP53* mutated cancers when compared to mock-treated animals. The third xenograft model of a *TP53* wildtype cancer did not respond to si*ENDOD1*. Our work opens up the possibility that a wide range of p53 mutations could be exploited using a SL approach to induce toxic DNA lesions specifically in cancer cells, but not in surrounding tissue. In summary, we identify ENDOD1 as a potential wide-spectrum and cancer-specific target for SL drug discovery.

## Methods

Information for cell lines, siRNA, primers and antibodies used in this study are listed in Supplementary Tables 1–4.

**Cell culture**. The $ENDOD1^{-/-}$ clone was obtained by targeting exon 1 with gRNA (5′-CAGCCTCTTCGCCCTGGCTGG-3′) in RPE1 cells (Supplementary Fig. 9c). The resulting insertion (C → CG) creates a frameshift in the ORF at amino acid position Arg4. All human cell lines were cultured in complete media supplemented with 10 or 20% foetal bovine serum (FBS) according to ATCC protocols. Where indicated, G1 arrest was induced by serum starvation for 8-11 days in complete media: RPE1 and $ENDOD1^{-/-}$, 1% FBS; GES-1, SKOV-3, NCI-H1299, MBA-MD-468 and C33A, 0.5% FBS. Representative FACS assays for cycling and serum starved cells are shown in Supplementary Fig. 9a. All cell lines tested negative for mycoplasma contamination and were authenticated by providers (National Collection of Authenticated Cell Cultures, Shanghai and iCell Bioscience Inc, Shanghai). Double stranded siRNA's were obtained from Ribobio, Guangzhou, China. Plasmid and siRNA transfections were performed using Lipofectamine 3000 (Invitrogen) or FuGENEHD transfection Reagent (Roche, E231A). 1 μg plasmid or 50 nM siRNA were applied per $10^6$ cells unless otherwise stated. Efficiencies of gene silencing for frequently used cell lines in this study are presented in Supplementary Fig. 10a–d. Targeting sequence of siRNA is listed in Supplementary Table 2.

**Plasmids**. For cloning of Flag-tagged *ENDOD1* and *TP53*, PCR fragments were inserted into the *Eco*RI site of pLVX-Flag -IRES-ZsGreen1 plasmid, using the In-Fusion cloning kit (Clontech, 639650). Point mutants for *ENDOD1* and *TP53* were converted from parental pLVX-Flag -IRES-ZsGreen1 plasmids using QuikChange Lightning Site-Directed Mutagenesis Kits (Stratagene, 200519). Truncations of *ENDOD1* and mutations of *TP53* were obtained by fusing different PCR fragments using In-Fusion cloning kit (Clontech, 639650). Primers used in this study were listed in Supplementary Table 3.

**Chemicals and genotoxic treatments**. To induce DNA damage, cells were treated with the indicated concentrations of: CPT (Selleck, S2423); HU (Selleck, S1896); Cisplatin (Supertrack Bio-pharmaceutical, 131102); CX5461 (Selleck, S2684) and $H_2O_2$ for the times stated for each experiment. Suppression of PARP enzymatic activities was achieved by adding pre-determined concentrations of Olaparib (Selleck, S1060) or Talaparib (Selleck, S7048) as indicated for individual experiments.

**LC-MS/MS for PARPi induced proteomic changes**. HBV-integrated T43 cells were continuously treated with 100 nM Olaparib for 2, 4 and 6 days. Whole cell samples were ground in liquid nitrogen and lysed in 8 M urea supplemented with 1% protease inhibitor cocktail (Calbiochem). Lysates were centrifuged at 12,000 × *g* and the supernatant was quantified by BCA assays (Beyotime). For mass spectrometry: in brief, protein solutions were reduced with dithiothreitol (5 mM) and alkylated with iodoace-tamide (11 mM). Samples were diluted by Triethyl ammonium bicarbonate to reduce the final concentration of urea to less than 2 M. After two rounds of trypsin (Promega) digestion, peptides were desalted using Strata X C18 SPE columns (Phenomenex) and reconstituted in 0.5 M TEAB. Peptide solutions were dissolved in UHPLC buffer A (0.1% (v/v) formic acid in water) and loaded for LC separation on a NanoElute high-performance liquid chromatography (UHPLC) system (Bruker Daltonics), using a 90 min LC gradient at 300 nL/min. MS data were collected using a tims-TOF Pro mass spectrometer (Bruker Daltonics), and processed using the Maxquant search engine (v.1.6.6.0). Mass spectra data were searched against SwissProt Human database concatenated with reverse decoy database.

**Cell proliferation and viability**. Cells treated with siRNA or chemicals were incubated in complete media. Proliferation curves were determined by counting cell number by haemocytometer every 3-5 days upon passage, dependent on the growth rate of individual cell lines. Inhibition rate (%) for each cell line was calculated as (1-siRNA/Control) × 100%. siRNA and control represent the number of remaining cells for the specific siRNA and si*Scramble* (si*Scr*.) control at the endpoint of the experiment. Relative cell survival: the ratio calculated by dividing the number of cells in the treatment group with untreated control at the endpoint of the experiment. For Giemsa staining, remaining cells were fixed with prechilled methanol for 10 min and stained with Giemsa solution (Baso, BA-4122). Cells subjected to CCK8 viability assay were replaced with fresh medium containing 10% CCK8 and incubated for 2–4 h. The colorigenic supernatant was carefully aspirated and transferred to new 96-well plates. Absorbance values at 450 nm was detected by Multifunctional enzyme marker. Relative cell survival was calculated according to OD values. A medium blank control was set for each plate and three replicates were included for statistical analysis.

**Measurement for PARP-DNA complex**. The method for assessing tight DNA/chromatin association of PARPs is described elsewhere[12]. Briefly, cells were treated for the indicated time with or without drugs and trypsinized. Pellets were then extracted with different stringency. 3 × $10^6$ cells were treated with 100 μl of hypotonic buffer (100 mM MES-NaOH, pH 6.4, 1 mM EDTA and 0.5 mM MgCl$_2$, 0.05% TritonX-100) supplemented with protease inhibitors (Complete Mini, Roche), layered gently onto 100 μl of hypotonic buffer containing 30% sucrose and centrifuged at 15,000 g at 4 °C for 10 min. The P1 fraction was derived by dissolving pellets in 100 μl of buffer A (50 mM HEPES-NaOH, pH 7.5, 100 mM KCl, 2.5 mM MgCl$_2$, 0.05% TritonX-100 and protease inhibitors). 50 μl of P1 was centrifuged at 15,000 g at 4 °C for 10 min. The supernatant was preserved (fraction A) and pellets dissolved in 50 μl buffer B (50 mM HEPES-NaOH, pH 7.5, 250 mM KCl, 2.5 mM MgCl$_2$, 0.05% TritonX-100 and protease inhibitors), centrifuged at 15,000 × g at 4 °C for 10 min. The supernatant was preserved (fraction B) and the pellet redissolved in 50 μl buffer C (50 mM HEPES-NaOH, ph 7.5, 500 mM KCl, 2.5 mM MgCl$_2$, 0.1% TritonX-100 and protease inhibitors), centrifuged at 15,000 g at 4 °C for 10 min. The supernatant was reserved (fraction C). The pellet was dissolved in 50 ml buffer A with 2 mM CaCl$_2$ and digested with micrococcal nuclease (M0247S, NEB) at RT for 20 min. The supernatant was collected (fraction D) after centrifugation at 15,000 × *g* at 4 °C for 10 min.

**Immunofluorescent staining**. Briefly, cells were grown on coverslips and fixed with Carnoy's fluid (methanol:glacial acid:3:1) or 4% paraformaldehyde (PFA) and permeabilize with 0.3% TritonX-100 followed by blocking in PBS with 3% BSA, 3% donkey serum and 0.2% Triton X-100. Primary antibodies were diluted with antibody buffer (0.1% Triton 5% BSA in PBS) and incubated for 2 h at ambient temperature. Primary antibodies were detected with anti-rabbit-Cy3 or anti-mouse-FITC. Fluorescent images were acquired using an Olympas (BX51) or LEICA DM4 B and images were processed analysed using Image-Pro Plus software. ENDOD1 Antibody used for immunofluoscent staining experiment is ABclonal (A16502). In general data from 200 to 500 cells were quantified per sample in each independent experiment for statistical analysis of imaging assays.

For non-denatured BrdU staining of ssDNA, proliferative cells on coverslips were pre-labelled with 40 ug/ml BrdU for 48 h before serum starvation, followed by fixation with methanol-acetic acid buffer (3:1) for 15 min. Coverslips were sequentially incubated in blocking buffer (0.3% Triton, 5% donkey serum in PBS) for 15 min and BrdU antibody (1:100) added and incubated overnight at 4 °C. Microscopic visualization and image capture were performed as described above.

TUNEL Cell death assays were performed following the instructions of DeadEnd™ Fluorometric TUNEL System kit (Promega, G3250). Briefly, cells were fixed in freshly prepared 4% methanol-free formaldehyde PBS solution for 25 min at 4 °C, followed by washing with PBS and 0.3% Triton X −100/PBS. After equilibration, cells were reacted with rTdT solution at 37 °C for 60 minutes and the reaction was terminated by adding 2 × SSC for 15 min at room temperature. Samples were re-stained with propidium iodide (1 μg/ml in PBS) in the dark and fluorescent images captured.

**Flow cytometry and fluorescent-activated cell sorting (FACS)**. Cell cycle analysis was performed using 3 × $10^4$ trypsin-dissociated cells. After rinsing in PBS two times cells were fixated with 75% ethanol overnight and stained with PBS-PI (50 μg/ml) for 20 min before cytometry using a BD FACSCalibur. Modfit software was used for data processing. For immunotyping of peripheral blood cells, ery-throcytes were removed by treatment with chilled RBC buffer (15 mM NH$_4$Cl, 1 mM KHCO$_3$, 0.1 mM EDTA, pH 7.1-7.4) for 10 min to cause lysis. The product was centrifuged for 5 min at 400 × *g* and cells (1–2 × $10^6$ cells/sample) were resuspended in 100 μl PBS, followed by incubation with the appropriate dilution of fluorescent antibody conjugates (listed in Supplementary Table 4) for 30 min at room temperature or 45 min on ice. For bone marrow analysis, a single cell suspension was obtained by flushing bone marrow cells with PBS containing 2% FBS, followed by incubating with marker antibody as above. Cells were considered as live cells after FSC/SSC gating and then used in fluorescence histograms. Labelled cells were analyzed on Beckman Cytoflex S and Flow Jo V10 software were used for

data analysis. Sorted cells were defined as the following: T cells (CD3e+), NK cells (NK1.1+), granulocyte (Gr-1+), macrophage (F480+ and CD11b+), hematopoietic stem cells (Lin- Sca1+ and c-Kit+). The border between negative and positive was determined by an isotype-matched control antibody. Gating strategy is exemplified in Supplementary Fig. 9b.

**Preparation of metaphase chromosomal spread**. Cells were plated in a 60-mm dish and arrested in mitosis by 2-h treatment with colcemid (final concentration; 200 ng/ml). Cells were trysinized and pre-warmed 0.075 M KCl added and incubated for 20 min at 37 °C. Four drops of freshly prepared fixative (3:1 solution of methanol:acetic acid) was added and cells were pelleted, resuspended in 5 ml fixative and incubated for 20 min at 4 °C. After repeating the fixation two times, pellets were resuspended in 0.5 ml fixative solution. Two or three drops of cell suspension were precipitated onto a pre-chilled microscope slide from a height of 18 inches. Slides were thoroughly air-dried and stained using Giemsa. The mitotic chromosomes were observed and evaluated using an Olympus fluorescence microscope (BX51) at ×1000 magnification.

**Comet assay**. Cells were treated with 10 mM $H_2O_2$ on ice and subsequently transferred to drug-free complete media for the indicated recovery periods. Cells were then trypsinized and resuspended in PBS and mixed with an equal volume of 1.6% low-gelling-temperature agarose (Thermo, 16520100) maintained at 42ºC. The mixture was immediately spread onto a frosted glass slide (Fisher) pre-coated with 0.8% agarose (Thermo, 16500500) and air dried over-night at RT to set. For alkaline electrophoresis, slides were immersed in pre-chilled lysis buffer (2.5 M NaCl, 10 mM Tris-HCl (pH10), 100 mM EDTA, 1% Triton X-100, 1% DMSO) for 1 h, washed twice with pre-chilled distilled water for 10 min and placed for 20 min in pre-chilled alkaline electrophoresis buffer (300 mM NaOH, 1 mM EDTA). Electrophoresis was then conducted at 0.75 V/cm for 15 min (0.75 V/cm for 30 min for detecting endogenous DNA breaks). Slides were and subsequently neutralized 3 times in 400 mM Tris-HCl pH7.0 for 5 min. For the neutral Comet Assay, electrophoresis solution (300 mM sodium acetate, 100 mM Tris-HCl pH 8.3) was used. DNA was fixed with absolute ethanol for 20 min then stained with EtBr (2 ug/ml) for 5 min and washed twice in ddH2O. The percentage of tail moment was calculated by dividing the intensity of tails by that of heads as measured with Image Pro Plus. Data presented in violin plots are 150 cells per data point from 3 independent biological repeats. The three repeats showed the same trend.

**S1 nuclease digestion**. Genomic DNA was extracted from $10^6$ cells using High Pure PCR Template Preparation Kit (Roche, 11796828001). 150 ng of purified DNA was digested in Reaction Buffer (200 mM sodium acetate [pH 4.5], 1.5 M NaCl and 10 mM ZnSO4) with different dilutions of S1 Nuclease for 10 mins at room temperature and inactivated by heating at 70 °C for 5 min in the presence of EDTA. Digested DNA was resolved by 0.8% agarose electrophoresis. Images were captured by Chemidoc XRS(Bio-Rad) and bands quantified by ImageJ.

**Animal, biopsies and histochemical staining**. The animal experiments in this study were registered and approved by the Medical Ethical Committee of the West China Second University Hospital of Sichuan University on 8 June 2018 (approval Reference number, Medical Research 2018 (015). Experiments were carried out in accordance with the approved guidelines. All mice were housed in standard SPF condition throughout the experiments at a maximum of 5 per cage with 12 h light/ dark cycles at 23 °C and 55% humidity.

C57BL/6JGpt mice were purchased from Gempharmatech co., Ltd. Sexually mature mice (C57/B6) were randomly divided into control siRNA treatment, sim*Endod1* and sim*Wdr70*. In vivo knockdown was performed by injecting specific double strand siRNA (10 mg/kg) and 100 μl RNA transfection reagent (Biotool, B45215). Injection started from 10 week after birth and occurred twice per weekly via alternative tail vein or intraperitoneal route. Body weight and the health condition of each mouse was examined weekly. Dissected tissues were embedded in paraffin and sliced at 8 μm for histochemical staining and images were captured by BX51 (Olympus). Five slices of biopsied tissues were examined for HE staining to avoid individual variation. Echocardiography was performed on a VisualSonics Vevo 2100 and output analysed with Vevostrain software, and operated by a person blinded to the treatment group. Animals were handled awake and held in a standard handgrip.

**Xenograft model in nude mice**. Female athymic nude immunodeficient mice (BALB/cGpt-Foxn1nu/Gpt, purchased from Gempharmatech co., Ltd) of 4–5 weeks of age were used for xenograft implants. Mice were subcutaneously inoculated in both sides of armpits or hind flanks ($2–4 \times 10^6$ cells *per* site). Animals were randomized using random number table into control and treatment groups when xenografts had reached an average volume of approximate 100 mm³. For treatment, animal was administered with 10 mg/kg *ENDOD1* siRNA twice a week with 100 μl RNA transfection reagent. A parallel group of mice was administered with control siRNA. Inoculated and siRNA-administered mice were observed each day. Tumour Size was measured by Vernier calliper and tumour volume calculated by the three-dimensional measurement (length × width × width/2) until termination of the experiments. Experiments were performed blind.

**Statistics**. All histograms are presented as means ± SEM. For quantitative analysis including immunoblotting, image analysis and repair analysis at least three independent biological repeats were carried out. Analysis for significant difference between two groups was performed either by Two-tailed Student's $t$ tests (2 sided) using GraphPad Prism 6 or SPSS 16.0. or by non-parametric Kruskal tests using Python scipy.stats package. End-point values of cell survival assays were used for statistical analysis. The level of significance was set as $p < 0.05$.

**Reporting summary**. Further information on research design is available in the Nature Research Reporting Summary linked to this article.

## Data availability

The data that support the findings of this study are available from the corresponding authors upon reasonable request. Source data are provided with the paper.

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

## Acknowledgements

This work is supported by NSF China (31771580, 82073027, 81702795, 81630038 and 81971433), Sichuan University (2020SCUNL204, SCU2019C4198), National Key Research and Development Program of China (2019YFE0198400, 2018YFC1002801), Department of Science and Technology of Sichuan Province (2019JDTD0020, 2020YFS0131, 2020YFH0019), and Wellcome Trust Award 110047/Z/15/Z. We thank Owen Wells, Penny Jeggo, Keith Caldecott, Dacai Liu and Jichen Hu for critical comments. We also thank PTM Biolab (Hangzhou) for mass spectrometry analysis.

## Author contributions

C.L., A.M.C. and Z.T. co-discovered a repair role for ENDOD1. C.L. and Z.T. established SL between ENDOD1, TP53 and HR defects, and formulated the SL model with A.M.C. M.Z. and X.W. performed biochemical and cell biology assays. P.Y. analysed heart function. C.G. and X.Z. assisted immunoblotting and cytometry. J.C., D.M., H.L. and D.K. advised on p53, animal experiments and DNA repair. C.L. and A.M.C. supervised the overall project and wrote the manuscript.

## Competing interests

A patent application 'Application of reagents or drugs inhibiting an endonuclease in cancer therapy (202010295776.6)' was filed on April 15, 2020.
