## [Peer Review File · Nature Communications]

REVIEWER COMMENTS

Reviewer #1 (Remarks to the Author):

This is an exciting and potentially very important paper allowing a new specific approach to the treatment of cancers based on loss of p53 function. The search for synthetic lethal interactions with p53 has prior to this study been unsuccessful. Here the authors present evidence that the gene ENDOD1 is synthetic lethal with Tp53 such that inhibition of ENDOD1 renders p53 mutant or null cells inviable while loss of ENDOD1 or loss/mutation of Tp53 is by itself not lethal. The observation is very exciting and reminds one of the breakthrough that has been made in the treatment of BRCA1/2 mutant breast cancers following the discovery that BRCA1/2 loss made breast cancer cells very sensitive to PARP inhibitors. The paper contains an immense amount of data and as it is constrained by the length requirements of the journal much of that data is held in the supplementary materials. Overall the data supports the concept of the SL interaction and establishes the utility of the process by showing the lack of toxicity to normal tissues of ENDOD1 knockdown. The mechanism of the synthetic lethality is partially established by showing that single strand DNA tracts are generated in the absence of p53 function and this requires the activity of the PARP enzyme. These single strand tracts are a unique feature of the p53 system since when HR genes are combined with loss of ENDOD1 DSB are generated but not ssDNA. Thus the paper implies a direct function for p53 in DNA repair acting to suppress ssDNA formation. The authors demonstrate the efficacy of ENDOD1 inhibition in a Xenograft model with impressive results. Here however they should include a control of a wild type p53 tumour showing that it is not sensitive to ENDOD1 inhibition. The synthetic lethal interaction does not require the cells to be in S phase implying that the repair process that p53 is involved in that suppresses the formation of ssDNA is not linked to replication. The paper as a whole is convincing but it could be improved by a simplification of the figures which are very dense and hard to analyse. Some of the labelling is incomplete. For example in the very first panel the most prominent bands on the western blot labeled with the * symbol are in fact "background" non specific bands but are not identified as such in the text or in the figure legend. It is only in the legend to one of the supplementary figures that the meaning of the * symbol is explained. I think the authors face a quandary due to the constraints of the journal and have used dense figures and text to try to explain a great deal in a short paper. For such an important finding it might actually be in their best interests to submit to a journal that can publish longer papers or simplify the amount of information given in this paper.

Reviewer #2 (Remarks to the Author):

Review of Tang et al., "Synthetic lethality between TP53 and ENDOD1"

In this rather disjointed and excessively terse manuscript, the authors first claim to have identified a truncated nuclear isoform of ENDOD1, then switch gears and investigate the effects of a global ENDOD1 knock-out on DNA damage, cell cycle progression, and tumor growth. They claim that RPE1 ENDOD1 -/- cells exhibit an apparent G1 repair defect, and are resistant to H2O2 treatment in a largely PARP-1 dependent manner. They then attempt to show, using siRNAs, that combined knock-down of many, but not all HR factors completely inhibits the growth of ENDOD1 knock-down cells, assuming that the

undocumented white and black bars in Fig S1e are +/- ENDOD1 knock-down.

Based on this observation, they then make a further leap and claim that ENDOD1 is synthetic lethal with p53 based on an analysis of five p53 mutant cell lines and two p53 wild-type cell lines, but a direct proof of p53 mutant dependency is never validated in an isogenic cell line that is + or - p53, such as HCT116 cell lines.

They next show using evidence of increased ssDNA in G1 cells that are ENDO1-/- following p53 knock-down using comet assays and phospho-RPA32 immunofluorescence, but it is not clear if these cells are truly isogenic with the RPE-1 cells shown for comparison, since they never restore wild-type ENDOD1 in the knock-out cell line, and the single clone they selected from the CRISPR targeting may be quite different from the bulk parental cell line. They go on to then complement SKOV-3 cells with wild-type or mutant p53 and demonstrate this enhanced PARylation and phospho-RPA foci formation with absent or mutant p53, confirming a synthetic interaction. Using siRNA against PARPs, or PARP inhibitors, they show that the PARP activity is required for the pRPA32 foci formation but again it is not clear if all the experiments were done in similar RPE1 cell lines.

They then attempt to correlate pRPA32 and PARP recruitment with knockdown of multiple HR factors, but do not comment that knockdown of several of these factors, all of which affect pRPA32 foci, but not PARylation (Fig. S7), does not correlate with the synthetic lethality seen when these same factors are knocked down (Fig S1e). They then jump to a mouse model and show what I guess may be some type of murine whole-body siRNA knock-down reagent (simEndod1, never defined) against ENDOD1, that the systemic treatment in mice is not toxic or myelosuppressive (Figure 4). They then jump to a murine xenograft model in which they show that C33A or SKOV3 tumor cells failed to show increased tumor growth as measured by tumor volume when mice were subject to full body siRNA mediated knockdown using simENDOD1 (which presumably is the same as the simEndod1 reagent used in the first part of Figure 4). Finally, the authors claim that this report is the first to describe evidence of “tumor-specific TP53 mutations could be treated via synthetic lethality”, a claim which is demonstrably false.

The basic observation from his paper - that ENDOD1 has a synthetic lethal interaction with p53, and that this is likely mediated by exposure of ssDNA in G1 cells, is extremely interesting, and worth more detailed exploration. Major problems with the current manuscript include the extraordinary terseness and limited explanation of any of the results, the lack of numerous critical controls, the poor description of what was actually done in each experiment, the complete absence of a molecular mechanism for the observations (which are, in fact, quite interesting), and the unsupported claim of first ever p53-mediated synthetic lethality. There are an extremely large number of problems with the manuscript in its current form. Some of the major criticisms that should be addressed include:

Major Comments:

1- The overall terseness and brevity with description of figures and results makes it extremely difficult understanding exactly what was done, and exactly what conclusions can confidently be drawn from the presented data. The introduction is limited to a short overview of the concept of synthetic lethality in the context of PARP inhibitors but fails to provide any background regarding ENDOD1, or why the authors were even interested in studying it. The known cellular roles of ENDOD1 and its relationship to DNA damage are unclear. It is also unclear as to how the authors found ENDOD1 as a protein to study. As best as we can tell, ENDOD1 was identified from a paper <https://doi.org/10.1016/j.celrep.2020.01.096> that investigated protein levels across different chromatin

states. This lack of attention to detail is evident throughout the manuscript, as exemplified, for example, by the following sentence in the first paragraph of the results. “siENDOD1 knockdown in GES-1 cells recapitulated H₂O₂ resistance and, in ENDOD1 ^{-/-} cells, resistance depended on PARP1 activity (Extended Data 1c)” Extended Data 1c does not have any data pertaining to PARP1. It does include treatment with CPT (topoisomerase poison), Cisplatin (cross-links), Cx5461 (G-quadruplex stabilization), HU (replication stress), IR (modified bases, single-strand breaks, double-strand breaks), and to H₂O₂, none of which are ever discussed or directly address PARP1 dependence.

2- A major part of the initial claim is that the authors identified a truncated nuclear isoform of ENDOD1. They attempt to characterize this truncated form via western blot including siRNA knockdown and CRISPR-genome knockout, but direct characterization of this form is completely lacking. Validation of the CRISPR knockout is also lacking as they show no sequencing trace of knockout and instead assert that there is single G insertion that causes an out of frame mutation (Fig S9b). The described G insertion is 31 bp away from the Cas9 cut site which is a highly improbable repair event, especially for a single bp insertion. Use of RPE-1 ENDOD1 ^{-/-} cells for a majority of the presented data should be met with proper validation of knockout status, assuming this knock-out truly was in RPE-1 cells, and the correct controls for a single cell ENDOD1 knock-out clone is that same clone +/- reconstitution with ENDOD1, not the parental heterogeneous cell population from which the KO was derived. The CRISPR KO that they have does show loss of the 55 Kd full-length as well as truncated ENDOD1 forms (Fig S1a). It would make a stronger case for the author's arguments if they also included multiple ENDOD1 KO clones. It is unclear if these are functional truncation products in cells or possible degradation products that occurred during analysis. Fig S1b analyzes Flag-tagged overexpression of ENDOD1 variants via western blot. Overexpression of the Flag-FL N-terminal tagged variant results in overexpression when probed with a ENDOD1 antibody (Fig S1b, left panel), however, there is an apparent lack of signal in the Flag blot (Fig S1b, right panel). There is no explanation or discussion of this likely signal sequence cleavage event. Overexpression of C-terminally tagged full-length ENDOD1 (Fig S1a, right blot) results in the appearance of multiple higher molecular weight species that are recognized by the ENDOD1 antibody. This observation is never noted or discussed even though it may be of significance to the story. Overexpression of Flag-22-334 (Fig S1b, left blot) also shows ENDOD1 intensity around the size of Flag-22-500 overexpression that migrates ~60 Kd. Why? This could perhaps be loading spillover or is that actually a species that is observed? Overall, the claim that the authors identified a truncated nuclear isoform of ENDOD1 is not accompanied by a rigorous characterization of the species, and nothing in the rest of the paper actually has anything to do with this specific truncated form.

3- siRNA mediated knockdown is utilized throughout the paper without proper validation of knockdown efficiency or specificity for any of the factors that are being targeted. The authors do show knockdown of individual PARPs, ENDOD1 or p53 in Fig S8 and S9. However, there is no validation of siRNA knock-downs for Fig 1d (PARP combinations), Fig S1e-f-g (ENDOD1, WDR70, BRCA1, BRCA2, ARID1A/1B, BLM, CtIP, CHK1, EXO1, MRE11, FANCA), Fig S2a-b-c-d-e (BRCA1, BRCA2, PARP1/2/3), Fig S3a-b (ENDOD1 in each cell line), etc. For each cell line that is used for siRNA mediated knockdown and in different conditions the authors should validate the knockdown of each protein ideally by western blot, but at least by qRT-PCR. In this respect, it seems especially surprising that siRNA knockdown of HR factors and ENDOD1 was apparently so lethal in Figure S1e, since most siRNA knock-downs are only partial. None of the experiments using siRNA knock-downs appear to have been done with multiple siRNAs, or had their

specificity directly examined by rescuing the lethality using re-expression of siRNA-resistant constructs.

4- One of the main proposed discoveries is that the truncated form of ENDOD1 increases its chromatin association following H₂O₂ treatment. Fig 1a, left panel shows an increase in the ~50 Kd “ND” fragment association with in the 250mM euchromatin condition but not in the 600mM heterochromatin condition. The authors do not discuss this difference observed between euchromatin and heterochromatin. It is also apparent that from Fig 1a, left panel that the full-length ENDOD1 protein shows increased euchromatin association after H₂O₂ treatment. The blot is somewhat overexposed making this observation harder to quantify, but still present. In the Fig 1a legend they state that Flag-22-344 ENDOD1 (~38 Kd) is expressed to show co-migration on the gel. However, this band migrates significantly above 38 Kd and closer to 50 Kd according to their marker reference. Either the marker reference is incorrectly labeled or the Flag-22-344 ENDOD1 species does not migrate at 38 Kd. Fig 1a, left panel should show a cytoplasm loading control to verify that it is a pure nuclear extract. The authors do nicely show the H3 (nuclear), c-MYC (euchromatin), and H3K27me3 (heterochromatin) blots but this does not show it is not contaminated with cytoplasmic components. Additionally, Fig1a right panel showing whole-cell lysates includes Hydroxyurea and CPT treatments. The reason for these treatments or the significance of them are never mentioned anywhere in the paper. Moreover, highlighting of increased chromatin association of this truncated nuclear isoform is never mentioned ever again. The authors quickly shift focus from the truncated nuclear isoform to the whole ENDOD1 knockout. It is unclear if any of the effects that are observed are due to the truncated nuclear isoform or full-length ENDOD1, both of which appear to increase chromatin association as mentioned before. The authors utilized two antibodies in Fig S1a/b during the initial characterization, however, it is unclear which antibody they used for further experiments throughout the paper, and proper characterization of the antibody specificity seems lacking.

5- Figure 1b analyzed immunofluorescence of PAR and ENDOD1 using RPE1 WT and RPE1 ENDOD1 KO cells. Given the intense non-specific bands that were present with both ENDOD1 polyclonal antibodies in Fig S1, immunofluorescence analysis might be difficult. However, the ENDOD1 ^{-/-} cells do not show any ENDOD1 staining under the conditions used as would be expected. It is not clear which antibody of the two they used for immunofluorescence. When comparing ENDOD1 staining in untreated vs 10mM H₂O₂ there appears to be a significant re-localization of ENDOD1 that is never addressed. The staining appears to be cytoplasmic in the untreated condition and a significant amount of nuclear localization happens after 10mM H₂O₂ treatment.

6- In Fig 1e the authors utilize multiple different cell lines with a range of TP53 status. The correlation between functional p53 and lack of synthetic interaction with ENDOD1 is clear. However, to make the claim of synthetic interaction between p53 and ENDOD1 the authors should utilize an isogenic cell line with reconstitution of p53 variants as they do in Fig 2f and Fig S5f. The observed growth differences may be attributed to the different genetic backgrounds of each cell line instead of being due to p53 status.

7- Throughout the paper the authors compare cell growth with cell count and relative cell number. A more thorough analysis of exactly what process is responsible for this difference in cell growth manifests should be performed. Is this difference occurring through slower cell growth or does it result from increased apoptotic or non-apoptotic cell death. The authors show that RPE1 ENDOD1 ^{-/-} cells do

accumulate in G1 in comparison to WT RPE1 cells. An investigation into the pathway of cell death would greatly reinforce the conclusions of the cell growth experiments as well as provide a clue to a possible mechanism.

8- The authors show that loss of ENDOD1 and TP53 results in increased single-stranded DNA as shown by both S1 endonuclease and pRPA32. These data in combination with the cell growth data hint at some interplay between ssDNA recognition resulting in arrested cell growth that is ablated with loss of p53. However, the authors do not present a clear, concise mechanism as to how this happens and is recognized by the cells. What is the interplay between the increased ssDNA and cell-cycle arrest with respect to ENDOD1? How exactly does ENDOD1 prevent ssDNA accumulation? Is this due to increased replication stress by improper entry into S-phase of ENDOD1^{-/-} TP53 KD cells or is it because there is some failure to complete break repair. Does ENDOD1 interact with PAR? How is ENDOD1 localized to chromatin? A more comprehensive analysis of the ssDNA phenomenon is needed to provide a clearer mechanism of the interaction between ENDOD1 and increased ssDNA.

9- All the experiments performed in the paper include siRNA and CRISPR mediated KD or KO of the proteins studied. Rescue experiments that restore expression of either p53 and/or ENDOD1 should be included to conclusively prove that the effects seen with synthetic lethality, ssDNA, pRPA, and increased PARylation are not due to off-target effects.

10- Mouse xenograft experiments involving characterization of SKOV3 and C33A and full-body targeting of ENDOD1 show that loss of ENDOD1 results in loss of SKOV3/C33A tumor growth. The use of siWDR70 in this experiment as well as characterization of myeloid/lymphoid/bone marrow HSC cells seems mostly tangential to the main message of the story. While this does show that whole-body knockdown of ENDOD1 does not result in gross changes in mouse health, it does not seem to add to showing this growth defect is due to ENDOD1. Use of whole-body knockout of ENDOD1 may have significant confounding factors. KD of ENDOD1 throughout the entire animal may significantly alter the tumor microenvironment and therefore the immune response to the implanted tumor cells.

11- The authors claim that this report is the first to describe evidence of “tumor-specific TP53 mutations could be treated via synthetic lethality”. I believe this has already been demonstrated for MAPKAP Kinase-2, in the setting of DNA damaging agents doi:10.1016/j.ccell.2015.09.009).

Other comments:

Fig 1b – PAR intensity graph should also include comparison between RPE1 WT and ENDOD1^{-/-} cells. Zoomed in figure of IF images to better show co-localization of PAR and ENDOD1 would. How many cells were included in the quantification?

Fig 1e – X-axis label needed

Fig 1e – Show non-merged images as DAPI merged images make BrdU staining differences hard to

visualize

Fig 2f – How many cells were used in quantification of foci?

Fig 3a - How many cells were used in quantification of foci?

Fig 4f – X-axis label needed

Fig S1a, right blot – some bands appear to be cropped off on the left side

Fig S1b, right blot – some bands appear to be cropped off right side

Fig S1d, cytoplasm fractionation blot

Fig S2a, b - # events quantified

Fig S2c, d – non-merged images so it's easier to see 53BP1 foci and # cells quantified

Fig S3a, b – x-axis label needed

Fig S6a, b – non-merged images and # events quantified

Fig S9c, d, e – MW markers

Reviewer #3 (Remarks to the Author):

The manuscript by Tang et al identifies a synthetic lethality interaction between TP53 and ENDOD1. The authors document this interaction in both cell lines and mouse models approaches. The findings have the potential for translational significance, since TP53 mutations are frequent in tumors. However, the manuscript brings little mechanistic insights to explain this interaction, and suffers from a lack of rigor making the results unconvincing.

Specific points:

1. The authors make no attempt to investigate the mechanism underlying the ENDOD1-p53 synthetic lethality. It is unclear what the role of p53 is in this context: is this a transcription-dependent or independent function? What are the relevant interactions/targets? How does p53-mediated cell cycle control and apoptosis fit into this? Similarly, how ENDOD1 acts it not addressed. How does it remove PARP1/PAR chains (this is suggested in the model figure). Not even simple experiments (eg if the nuclease activity is involved) are presented. Without a solid understanding of the roles of ENDOD1 and 53 which are underlying the synthetic lethality, the manuscript cannot be published in Nature Communications.

2. Throughout the manuscript, the authors are employing a single siRNA for both ENDOD1 and TP53. This is a significant weakness, as off target effects for siRNA oligonucleotides are very frequent. (While the authors also employ an ENDOD1-knockout cell line as well as cell lines with p53 mutations, those lines are in general used for a different set of experiments -see comment below). Basically, none of the experiments which are done with siRNA alone can be trusted if only one oligonucleotide is used. This includes all the proliferation experiments. The authors need to perform all these experiments using multiple siRNAs.

3. In the same vein: the authors employ a single ENDOD1-knockout clone. As this was obtained by CRISPR, to rule out off-target effects the authors need to employ multiple independent knockout clones, or at the very least complement the knockout clone at hand by re-expressing exogenous ENDOD1.

4. The authors employ ENDOD1 siRNA and knockout approaches throughout the manuscript in a seemingly random manner. For example, the comet and drug sensitivity assays are done with the knockout clone, and the proliferation experiments are done with siRNA. What is the rationale for this approach? All important experiments should be performed with both knockdown and knockout.

5. Another major weakness is the use of “proliferation” experiments to measure synthetic lethality. The authors employed hemocytometer counting for these experiments, but this is an inaccurate method. Since the differences reported are minor (eg Fig 1d, 2a), these results are not convincing. The authors need to rigorously measure survival/proliferation using a variety of different assays, including clonogenic assays and measurement of cell proliferation using MTT, CellTiterGlo or similar reagents.

6. On page 3 when discussing fig 2, the authors state that “concomitant loss of p53 and ENDOD1 results in de-novo generation of ssDNA”. What does “de novo” mean in this context? Do the authors imply that this occurs during DNA replication? This would in fact be an important point since it may allow some mechanistic insights, so the authors should address it by labeling cells with BrdU and investigating ssDNA formation in BrdU-labeled cells.

7. Does loss of PARP1 restore DNA resistance to S1 nuclease in the P53-ENDOD1 codepleted cells? This would be important to assess, in order to establish the ssDNA accumulation as the cause of the synthetic lethality.

8. What is the relevance of the “truncated nuclear isoform” presented in figure 1? No studies are presented to identify the exact nature of the band on the blot, and if it is functionally relevant. Thus it makes no sense to introduce and discuss it. The authors write in the third line of the Discussion section that they “identify an additional role for a nuclear isoform of ENDOD1 in regulating PARP” but there is absolutely no evidence of this presented. The immunofluorescence experiment in Fig 1b is performed with an antibody recognizing the full length ENDOD1, so it cannot be implied that the co-localization described is with the truncation.

9. Fig 2b does not seem to present the quantification of the colocalization between PAR and ENDOD1, but rather the quantifications of each of them separately. Just because they follow the same trend, it

cannot be concluded that they co-localize.

10. The statistical analyses for the comet experiments (eg Fig 1c, 2c) are missing. How many independent experiments were performed? How were the data analyzed to ensure that the effects observed are statistically significant?

11. In order to address the clinical relevance of these findings, the authors should explore TCGA and other similar databases for evidence of this synthetic lethality. For example, one would anticipate that mutations in TP53 and ENDOD1 are mutually exclusive in tumors. Is this the case?

12. The model in Fig 3g is not described anywhere, not in the text nor in the figure legend. It is unclear how the model is supported by the data presented, or if this is a purely fictional model.

13. The manuscript is written in a very concise manner, which is not beneficial to the paper. It is unclear why each of the experiments were performed, what was the rationale, and what was gained from that particular experiment. The reader is left to guess the take home message of each section. No logical transition from one section to the other is presented.

We thank the reviewers for their helpful comments. We have made a significant number of changes and added data to address their comments. Specifically (and as detailed below in the point-by-point responses) we have included control data for knockdown efficacy, we have added additional mechanistic data and we have completely re-written the manuscript, expanding on the previous iteration with additional explanations.

Reviewer #1

This is an exciting and potentially very important paper allowing a new specific approach to the treatment of cancers based on loss of p53 function. The search for synthetic lethal interactions with p53 has prior to this study been unsuccessful. Here the authors present evidence that the gene ENDOD1 is synthetic lethal with Tp53 such that inhibition of ENDOD1 renders p53 mutant or null cells inviable while loss of ENDOD1 or loss/mutation of Tp53 is by itself not lethal. The observation is very exciting and reminds one of the breakthrough that has been made in the treatment of BRCA1/2 mutant breast cancers following the discovery that BRCA1/2 loss made breast cancer cells very sensitive to PARP inhibitors. The paper contains an immense amount of data and as it is constrained by the length requirements of the journal much of that data is held in the supplementary materials. Overall the data supports the concept of the SL interaction and establishes the utility of the process by showing the lack of toxicity to normal tissues of ENDOD1 knockdown. The mechanism of the synthetic lethality is partially established by showing that single strand DNA tracts are generated in the absence of p53 function and this requires the activity of the PARP enzyme. These single strand tracts are a unique feature of the p53 system since when HR genes are combined with loss of ENDOD1 DSB are generated but not ssDNA. Thus the paper implies a direct function for p53 in DNA repair acting to suppress ssDNA formation. The authors demonstrate the efficacy of ENDOD1 inhibition in a Xenograft model with impressive results. Here however they should include a control of a wild type p53 tumour showing that it is not sensitive to ENDOD1 inhibition.

We agree that a control p53 wildtype tumour Xenograft is an important point and thus established a third xenograft model using the *TP53* wildtype cancer line (MDA-MB-361). Consistent with our expectations, this control tumour did not respond to si*ENDOD1* (Supplementary Figure 8d).

*The synthetic lethal interaction does not require the cells to be in S phase implying that the repair process that p53 is involved in that suppresses the formation of ssDNA is not linked to replication. The paper as a whole is convincing but it could be improved by a simplification of the figures which are very dense and hard to analyse. Some of the labelling is incomplete. For example in the very first panel the most prominent bands on the western blot labeled with the * symbol are in fact "background" non specific bands but are not identified as such in the text or in the figure legend. It is only in the legend to one of the supplementary figures that the meaning of the * symbol is explained. I think the authors face a quandary due to the constraints of the journal and have used dense figures and text to try to explain a great deal in a short paper. For such an important finding it might actually be in their best interests to submit to a journal that can publish longer papers or simplify the amount of information given in this paper.*

We apologise for brevity and the incomplete descriptions in the text and figure legend. In the revised version we have tried to explain the logic of our approach and interpretation of each phenomenon observed. We also have also tried to simplify the figure presentation and pay attention to the labelling.

Reviewer #2

Review of Tang et al., "Synthetic lethality between TP53 and ENDOD1"
In this rather disjointed and excessively terse manuscript, the authors first claim to have identified a truncated nuclear isoform of ENDOD1, then switch gears and investigate the effects of a global ENDOD1 knock-out on DNA damage, cell cycle progression, and tumor growth. They claim that

RPE1 ENDOD1^{-/-} cells exhibit an apparent G1 repair defect, and are resistant to H₂O₂ treatment in a largely PARP-1 dependent manner. They then attempt to show, using siRNAs, that combined knock-down of many, but not all HR factors completely inhibits the growth of ENDOD1 knock-down cells, assuming that the undocumented white and black bars in Fig S1e are +/- ENDOD1 knock-down.

Based on this observation, they then make a further leap and claim that ENDOD1 is synthetic lethal with p53 based on an analysis of five p53 mutant cell lines and two p53 wild-type cell lines, but a direct proof of p53 mutant dependency is never validated in an isogenic cell line that is + or - p53, such as HCT116 cell lines. They next show using evidence of increased ssDNA in G1 cells that are ENDOD1^{-/-} following p53 knock-down using comet assays and phospho-RPA32 immunofluorescence, but it is not clear if these cells are truly isogenic with the RPE-1 cells shown for comparison, since they never restore wild-type ENDOD1 in the knock-out cell line, and the single clone they selected from the CRISPR targeting may be quite different from the bulk parental cell line. They go on to then complement SKOV-3 cells with wild-type or mutant p53 and demonstrate this enhanced PARylation and phospho-RPA foci formation with absent or mutant p53, confirming a synthetic interaction. Using siRNA against PARPs, or PARP inhibitors, they show that the PARP activity is required for the pRPA32 foci formation but again it is not clear if all the experiments were done in similar RPE1 cell lines.

They then attempt to correlate pRPA32 and PARP recruitment with knockdown of multiple HR factors, but do not comment that knockdown of several of these factors, all of which affect pRPA32 foci, but not PARylation (Fig. S7), does not correlate with the synthetic lethality seen when these same factors are knocked down (Fig S1e). They then jump to a mouse model and show what I guess may be some type of murine whole-body siRNA knock-down reagent (simEndod1, never defined) against ENDOD1, that the systemic treatment in mice is not toxic or myelosuppressive (Figure 4). They then jump to a murine xenograft model in which they show that C33A or SKOV3 tumor cells failed to show increased tumor growth as measured by tumor volume when mice were subject to full body siRNA mediated knockdown using simENDOD1 (which presumably is the same as the simEndod1 reagent used in the first part of Figure 4). Finally, the authors claim that this report is the first to describe evidence of "tumor-specific TP53 mutations could be treated via synthetic lethality", a claim which is demonstrably false. The basic observation from his paper - that ENDOD1 has a synthetic lethal interaction with p53, and that this is likely mediated by exposure of ssDNA in G1 cells, is extremely interesting, and worth more detailed exploration. Major problems with the current manuscript include the extraordinary terseness and limited explanation of any of the results, the lack of numerous critical controls, the poor description of what was actually done in each experiment, the complete absence of a molecular mechanism for the observations (which are, in fact, quite interesting), and the unsupported claim of first ever p53-mediated synthetic lethality. There are an extremely large number of problems with the manuscript in its current form. Some of the major criticisms that should be addressed include:

We agree the style of presentation in our initial submission was 'excessively terse'. We have significantly expanded the description of the data and the logic leading up to specific experiments and hope the referee finds this easier to follow. We have also attempted to improve the layout of figures and data labelling.

Specific comments as above:

A. They then attempt to show, using siRNAs, that combined knock-down of many, but not all HR factors completely inhibits the growth of ENDOD1 knock-down cells, assuming that the undocumented white and black bars in Fig S1e are +/- ENDOD1 knock-down

We repeated the experiment for ENDOD1-HRD SL using two different sets of siRNAs. The updated results (Figure 2a) show that silencing of all HR factors tested in the *ENDOD1^{-/-}* background cause growth defects. The results for siMRE11 and siBLM now show sensitivity, different from the initial figure. This is likely due to the lack of efficacy of the initial reagents used.

B. Based on this observation, they then make a further leap and claim that ENDOD1 is synthetic lethal with p53 based on an analysis of five p53 mutant cell lines and two p53 wild-type cell lines, but a direct proof of p53 mutant dependency is never validated in an isogenic cell line that is + or - p53, such as HCT116 cell lines.

We apologise for the brief nature of the first submission. It was not the case that the ENDOD1-TP53 SL was based on observations in Figure 1. We should have explained more clearly that it was in fact identified serendipitously based on a new observation – the majority of cell lines that we tested were unexpectedly sensitive to ENDOD1 ablation and many were not HRD. In Figure 1f (original version), the ENDOD1-TP53 SL analysis included 15 TP53 mutant and 6 TP53 + lines (not 5 and 2 as stated above). We agree with referee's advice that this should have been validated in an isogenic cell system. We investigated the toxicity of ENDOD1 ablation using coisogenic HCT116 cells: HCT116 cells expressing wildtype TP53 maintained proliferation, but those expressing mutated TP53 did not proliferate (see Supplementary Figure 4h). This is fully consistent with the experiment (now shown as Figure 4g) where we complement p53-null SKOV3 cells with wild type and mutant p53 and showed that only wild type and LW22/23QS p53 prevent the formation of RPA foci.

C. They next show using evidence of increased ssDNA in G1 cells that are ENDO1-/- following p53 knock-down using comet assays and phospho-RPA32 immunofluorescence, but it is not clear if these cells are truly isogenic with the RPE-1 cells shown for comparison, since they never restore wild-type ENDOD1 in the knock-out cell line, and the single clone they selected from the CRISPR targeting may be quite different from the bulk parental cell line.

We agree with the referee's view that the isogenic nature of ENDOD1-/- and RPE1 should be addressed. We reintroduced ENDOD1 by lentivirus infection and evaluated the viability of ENDOD1-/- after siTP53 or not. Infection of virus expressing ENDOD1, but not empty virus, rescued ENDOD1-TP53 SL (Supplementary Figure 4f). The ssDNA generation in ENDOD1-/- siTP53 cells could also be rescued by ectopic expression of ENDOD1 (Supplementary Figure 5d). This is fully consistent with the observation (see above) that the ENDOD1-TP53 SL is also validated in coisogenic HCT119 lines expressing wildtype or mutant TP53 and with our reconstitution of p53 (mutant or wildtype) in SKOV3 cells.

D. Using siRNA against PARPs, or PARP inhibitors, they show that the PARP activity is required for the pRPA32 foci formation but again it is not clear if all the experiments were done in similar RPE1 cell lines.

We apologise for inadequate description of the experiments in previous the version.

E. They then attempt to correlate pRPA32 and PARP recruitment with knockdown of multiple HR factors, but do not comment that knockdown of several of these factors, all of which affect pRPA32 foci, but not PARylation (Fig. S7), does not correlate with the synthetic lethality seen when these same factors are knocked down (Fig S1e).

Hopefully our more verbose presentation now makes our logic clear. In the new version we have expanded our mechanistic work and now show (Figure 6) that XRCC1 is required for PAR catalysation by PARP when ENDOD1 and p53 are simultaneously absent. We also show that the impact of XRCC1 acts upstream of the resection machinery, since the ablation

of *XRCC1* significantly reduced levels of MRE11 staining (see Supplementary Figure 7d). Consistent with this, in the absence of resection factor, RPA signal disappears but PAR is unaffected – i.e. PAR catalysation by PARP is dependent on PARP1-*XRCC1*, but not resection factors.

We also discuss in detail that *ENDOD1*-HRD and *ENDOD1*-TP53 are two distinct types of SL. It is conceivable that HR factors play different roles in the context of the two different synthetic lethality, considering the fact that HR defects mainly affect DNA repair in S/G2, while ssDNA production in *ENDOD1*-TP53 double mutant occurs also in G1 phase. In the following Figure, we show that knockdown of *BRCA1* alleviates the lethality of G1-arrested *ENDOD1*^{-/-} siTP53 cells besides blocking ssDNA generation (Figure 6d). In contrast, these treatments cannot rescue, indeed they further reduce, the viability of *ENDOD1*^{-/-} siTP53 cells when they are cycling. This ‘discrepancies’ are likely due to the compound impact of the two distinct synthetic lethality: *ENDOD1*-HRD and *ENDOD1*-TP53. Address in this awaits further characterisation and is outside the scope of the current work.

F. They then jump to a mouse model and show what I guess may be some type of murine whole-body siRNA knock-down reagent (simEndod1, never defined) against ENDOD1, that the systemic treatment in mice is not toxic or myelosuppressive (Figure 4).

Hopefully, in the new version we have explained better why we used a mouse model to address this issue.

G. They then jump to a murine xenograft model in which they show that C33A or SKOV3 tumor cells failed to show increased tumor growth as measured by tumor volume when mice were subject to full body siRNA mediated knockdown using simENDOD1 (which presumably is the same as the simEndod1 reagent used in the first part of Figure 4)

We apologise for the mislabelling of the siRNA's mEndod1 (murine) and ENDOD1 (human) in initial version, which would understandably confuse the reader. Corrections are made in the new version.

H. Finally, the authors claim that this report is the first to describe evidence of “tumor-specific TP53 mutations could be treated via synthetic lethality”, a claim which is demonstrably false.

We thank the referee for drawing our attention to this paper. We agree that previous work has revealed synthetic sensitivity between MAPKAP Kinase-2 and p53 in the context of chemotherapeutic treatments. We now describe our finding as: “These data identify ENDOD1 as a novel cancer-specific target for SL drug discovery.”

Major Comments:

1- The overall terseness and brevity with description of figures and results makes it extremely difficult understanding exactly what was done, and exactly what conclusions can confidently be drawn from the presented data. The introduction is limited to a short overview of the concept of synthetic lethality in the context of PARP inhibitors but fails to provide any background regarding ENDOD1, or why the authors were even interested in studying it. The known cellular roles of ENDOD1 and its relationship to DNA damage are unclear. It is also unclear as to how the authors found ENDOD1 as a protein to study. As best as we can tell, ENDOD1 was identified from a paper <https://doi.org/10.1016/j.celrep.2020.01.096> that investigated protein levels across different chromatin states. This lack of attention to detail is evident throughout the manuscript, as exemplified, for example, by the following sentence in the first paragraph of the results. “siENDOD1 knockdown in GES-1 cells recapitulated H2O2 resistance and, in ENDOD1 -/- cells, resistance depended on PARP1 activity (Extended Data 1c)” Extended Data 1c does not have any data pertaining to PARP1. It does include treatment with CPT (topoisomerase poison), Cisplatin (cross-links), Cx5461 (G-quadruplex stabilization), HU (replication stress), IR (modified bases, single-strand breaks, double-strand breaks), and to H2O2, none of which are ever discussed or directly address PARP1 dependence.

Again, we apologize for the terseness and brevity of last version and hope we have improved the experimental descriptions and logic flow in current manuscript. We also elaborate the sparse background of ENDOD1 and explain why we started studying it because it initially appeared to be regulated by Cul4-WDR70 (mass spectrometry experiments) in response to PARP inhibitor (Supplementary Figure 1a). By the time we had demonstrated that it was not, in fact, under control of Cul4, we had accumulated the preliminary data suggesting it had a role in response to H2O2 treatment that was influenced by PARP status and thus we continued exploring its function.

ENDOD1 has not been intensively studied in the literature. Initially ENDOD1 was characterised as a putative nuclease in fish (see PMID: 26784919). The only significant experimental report regarding human ENDOD1 suggests that it functions in innate immunity as part of the cGAS-STING pathway (Fenech et al., 2020 PMID: 32614325). Other publications fail to provide any data to clarify its function, although Federation et al did describe ENDOD1 as being present in a chromatin fraction and several sporadic reports have associated to locus with various cancers (i.e. PMID: 28532481, 21829708, 32899691, 32977589, 20165692). However, we are unable to make a coherent model from these association studies. We also apologize for the mislabelling of “Extended Data 1c”. We describe the PARP1-dependent H2O2 resistance in ENDOD1-/- cells in the revised version in Figure 1d and Supplementary Figure 2d.

2- A major part of the initial claim is that the authors identified a truncated nuclear isoform of ENDOD1. They attempt to characterize this truncated form via western blot including siRNA knockdown and CRISPR-genome knockout, but direct characterization of this form is completely lacking.

We agree with referee’s opinion that the truncated nuclear form is under-characterized, which is partly due to its scarcity in cells. To avoid confusion during reading, we removed this data set and its description from the current manuscript and focus on the repair and SL phenotypes observed upon ENDOD1 ablation.

Validation of the CRISPR knockout is also lacking as they show no sequencing trace of knockout and instead assert that there is single G insertion that causes an out of frame mutation (Fig S9b). The described G insertion is 31 bp away from the Cas9 cut site which is a highly improbable repair event, especially for a single bp insertion. Use of RPE-1 ENDOD1 -/- cells for a majority of the presented data should be met with proper validation of knockout status, assuming this knock-out truly was in RPE-1 cells, and the correct controls for a single cell ENDOD1 knock-out clone is that same clone +/- reconstitution with ENDOD1, not the parental heterogeneous cell population from which the KO was derived. The CRISPR KO that they have does show loss of the 55 Kd full-length as well as truncated ENDOD1 forms (Fig S1a). It would make a stronger case for the author's arguments if they also included multiple ENDOD1 KO clones.

As stated above we are also cautious concerning potential genetic variation between RPE1 and ENDOD1-/- . Following the referee's request, we provide the sequence trace showing the homozygous G insertion. We only isolated one ENDOD1-/- clone and did not obtain any ENDOD1 +/- clones from the KO screen. We cannot explain the reason why this insertion occurred 31 bp from Cas9 site. We have validated the utility of the ENDOD1-/- cell line by reconstitution (as explained above), and also performed a large number of control or verification experiments, including those performed with different siRNAs (Figure 1d, 2a, 2c, 3c, 3d, 5c, 5e, 6d, Supplementary Figure 3a, 3b, 4b, 4h and 5c), complementation for the survival and signal generation in ENDOD1-TP53 deficient cells (Supplementary Figure 4f and 5d for ENDOD1-/-, and Supplementary Figure 6c-e for SKOV3), as well as reproducing experiments in different cell lines throughout the paper. All of these data support the conclusion that ENDOD1-TP53 SL is caused by specific genetic interaction rather than being the outcome of random mutations.

It is unclear if these are functional truncation products in cells or possible degradation products that occurred during analysis. Fig S1b analyzes Flag-tagged overexpression of ENDOD1 variants via western blot. Overexpression of the Flag-FL N-terminal tagged variant results in overexpression when probed with a ENDOD1 antibody (Fig S1b, left panel), however, there is an apparent lack of signal in the Flag blot (Fig S1b, right panel). There is no explanation or discussion of this likely signal sequence cleavage event.

We agree that we cannot explain the multiple bands seen for ENDOD1 in immunoblotting. These bands appear in different cell lines (Supplementary Figure 10d) and are revealed by two different antibodies (Figure 1b,c). At this stage, we can conclude that the multiple bands represent natural isoforms (likely truncation) rather than non-specific signals, but their biological functions are currently not understood. We also discuss in Figure 1a and in the text the N-terminal signal peptide (residues 1-22), the cleavage of which likely accounts for the different signal detected by N- and C-terminal tagged ENDOD1: "This suggests that the signal peptide is cleaved as predicted".

Overexpression of C-terminally tagged full-length ENDOD1 (Fig S1a, right blot) results in the appearance of multiple higher molecular weight species that are recognized by the ENDOD1 antibody. This observation is never noted or discussed even though it may be of significance to the story. Overexpression of Flag-22-334 (Fig S1b, left blot) also shows ENDOD1 intensity around the size of Flag-22-500 overexpression that migrates ~60 Kd. Why? This could perhaps be loading spillover or is that actually a species that is observed? Overall, the claim that the authors identified a truncated nuclear isoform of ENDOD1 is not accompanied by a rigorous characterization of the species, and nothing in the rest of the paper actually has anything to do with this specific truncated form.

We have investigated the HMW species and showed that K48/K63 chains are present in these bands. However, this is not particularly informative and we do not feel that it is appropriate at this time to include this data in current paper.

3- siRNA mediated knockdown is utilized throughout the paper without proper validation of knockdown efficiency or specificity for any of the factors that are being targeted. The authors do show knockdown of individual PARPs, ENDOD1 or p53 in Fig S8 and S9. However, there is no validation of siRNA knock-downs for Fig 1d (PARP combinations), Fig S1e-f-g (ENDOD1, WDR70, BRCA1, BRCA2, ARID1A/1B, BLM, CtIP, CHK1, EXO1, MRE11, FANCA), Fig S2a-b-c-d-e (BRCA1, BRCA2, PARP1/2/3), Fig S3a-b (ENDOD1 in each cell line), etc. For each cell line that is used for siRNA mediated knockdown and in different conditions the authors should validate the knockdown of each protein ideally by western blot, but at least by Qrt-PCR. In this respect, it seems especially surprising that siRNA knockdown of HR factors and ENDOD1 was apparently so lethal in Figure S1e, since most siRNA knock-downs are only partial. None of the experiments using siRNA knock-downs appear to have been done with multiple siRNAs, or had their specificity directly examined by rescuing the lethality using re-expression of siRNA-resistant constructs.

For the validation of knockdown efficiencies, we provided immunoblotting or semi-quantitative PCR for relevant knockdown experiments in Figures 1d, 2c, 5c, 5h and Supplementary Figures 3a, 3b, 3e and 10. Due to the length restriction, we only include representative knockdown assessment for key experiments. Given the lethality of complete ablation of single HR factors, it is ideal not to knockdown these gene extensively, otherwise it would be difficult to evaluate the SL effect between ENDOD1 and HR factors. In the context of SL, we do not find it surprising that “*siRNA knockdown of HR factors and ENDOD1 was apparently so lethal in Figure S1e, since most siRNA knock-downs are only partial*”, a concern expressed by the referee. We have, however, repeated key experiments and performed new assays with different siRNAs. Data are shown in Figure 1d, 2a, 2c, 3c, 3d, 5c, 5e, 6d, Supplementary Figure 3a, 3b, 4b, 4h and 5c.

4- One of the main proposed discoveries is that the truncated form of ENDOD1 increases its chromatin association following H2O2 treatment. Fig 1a, left panel shows an increase in the ~50 Kd “ND” fragment association with in the 250mM euchromatin condition but not in the 600mM heterochromatin condition. The authors do not discuss this difference observed between euchromatin and heterochromatin. It is also apparent that from Fig 1a, left panel that the full-length ENDOD1 protein shows increased euchromatin association after H2O2 treatment. The blot is somewhat overexposed making this observation harder to quantify, but still present. In the Fig 1a legend they state that Flag-22-344 ENDOD1 (~38 Kd) is expressed to show co-migration on the gel. However, this band migrates significantly above 38 Kd and closer to 50 Kd according to their marker reference. Either the marker reference is incorrectly labeled or the Flag-22-334 ENDOD1 species does not migrate at 38 Kd. Fig 1a, left panel should show a cytoplasm loading control to verify that it is a pure nuclear extract. The authors do nicely show the H3 (nuclear), c-MYC (euchromatin), and H3K27me3 (heterochromatin) blots but this does not show it is not contaminated with cytoplasmic components. Additionally, Fig1a right panel showing whole-cell lysates includes Hydroxyurea and CPT treatments. The reason for these treatments or the significance of them are never mentioned anywhere in the paper. Moreover, highlighting of increased chromatin association of this truncated nuclear isoform is never mentioned ever again. The authors quickly shift focus from the truncated nuclear isoform to the whole ENDOD1 knockout. It is unclear if any of the effects that are observed are due to the truncated nuclear isoform or full-length ENDOD1, both of which appear to increase chromatin association as mentioned before. The authors utilized two antibodies in Fig S1a/b during the initial characterization, however, it is unclear which antibody they used for further experiments throughout the paper, and proper characterization of the antibody specificity seems lacking.

As we stated in the response to question 2, the potential protein isoforms of ENDOD1 are under-investigated, and chromatin fractionation data in original Figure 1a have been withdrawn to focus on the phenotypic consequences of ENDOD1 ablation. The use of ENDOD1 antibodies is specified in Figure legends and Supplementary Methods.

5- Figure 1b analyzed immunofluorescence of PAR and ENDOD1 using RPE1 WT and RPE1 ENDOD1 KO cells. Given the intense non-specific bands that were present with both ENDOD1 polyclonal antibodies in Fig S1, immunofluorescence analysis might be difficult. However, the ENDOD1 -/- cells do not show any ENDOD1 staining under the conditions used as would be expected. It is not clear which antibody of the two they used for immunofluorescence. When comparing ENDOD1 staining in untreated vs 10mM H2O2 there appears to be a significant re-localization of ENDOD1 that is never addressed. The staining appears to be cytoplasmic in the untreated condition and a significant amount of nuclear localization happens after 10mM H2O2 treatment.

The concerns about staining specificity are likely explained by methanol fixation, which washes away many soluble proteins that may include the non-specific bands detected in immunoblotting. To address referee's concerns on subcellular localisation, we stated in the updated version: "*While ENDOD1 was first identified as a cytoplasmic protein, a recent mass spectrometry study identified ENDOD1 peptides in the nucleus. In the context of our identification of damage-induced intra-nuclear ENDOD1 foci, this is consistent with an additional role for ENDOD1 in DNA repair*". This makes clear that ENDOD1 has functions in both in the cytoplasm, as discovered before, and in damaged nuclei as we show in this study.

6- In Fig 1e the authors utilize multiple different cell lines with a range of TP53 status. The correlation between functional p53 and lack of synthetic interaction with ENDOD1 is clear. However, to make the claim of synthetic interaction between p53 and ENDOD1 the authors should utilize an isogenic cell line with reconstitution of p53 variants as they do in Fig 2f and Fig S5f. The observed growth differences may be attributed to the different genetic backgrounds of each cell line instead of being due to p53 status.

We agree that our claim about SL between ENDOD1 and TP53 is strengthened by the use of coisogenic cell lines and thus we have, as suggested, extended our use of these to show that HCT116 lines carrying mutant TP53 but not the one carrying wildtype TP53 is sensitive to ENDOD1 ablation (Supplementary Figure 4h)

7- Throughout the paper the authors compare cell growth with cell count and relative cell number. A more thorough analysis of exactly what process is responsible for this difference in cell growth manifests should be performed. Is this difference occurring through slower cell growth or does it result from increased apoptotic or non-apoptotic cell death. The authors show that RPE1 ENDOD1 -/- cells do accumulate in G1 in comparison to WT RPE1 cells. An investigation into the pathway of cell death would greatly reinforce the conclusions of the cell growth experiments as well as provide a clue to a possible mechanism.

The manner of cell death is indeed intriguing question. We show (old version supplementary Figure S4c, new version Supplementary Figure 4d) that ENDOD1-/- cells die by apoptosis. We do not feel that a more extensive characterisation of cell death in the multiple cell lines used here is directly relevant at this point. We have repeated many experiments using CCK8 viability assay to replace cell counting.

8- The authors show that loss of ENDOD1 and TP53 results in increased single-stranded DNA as shown by both S1 endonuclease and pRPA32. These data in combination with the cell growth data hint at some interplay between ssDNA recognition resulting in arrested cell growth that is ablated with loss of p53. However, the authors do not present a clear, concise mechanism as to how this happens and is recognized by the cells. What is the interplay between the increased ssDNA and cell-cycle arrest with respect to ENDOD1?

While we agree there is a lot of mechanistic detail required to fully understand the phenomenon we observe, we respectfully suggest much of what the referee requests is outside of the scope of the current manuscript.

How exactly does ENDOD1 prevent ssDNA accumulation? Is this due to increased replication stress by improper entry into S-phase of ENDOD1-/- TP53 KD cells or is it because there is some failure to complete break repair. Does ENDOD1 interact with PAR? How is ENDOD1 localized to chromatin? A more comprehensive analysis of the ssDNA phenomenon is needed to provide a clearer mechanism of the interaction between ENDOD1 and increased ssDNA.

We assayed the G1-arrested cells to ensure that they do not go illegitimately into bulk DNA replication. As shown in Supplementary Figure 4g, we did not detect any DNA replication (by BrdU pulse-chase and denatured staining) in either RPE1 or ENDOD1-/- cells. We favour the referee's second proposal (*failure to complete break repair*). Our new data reinforces this interpretation: we show in Figure 6 that ssDNA production and PAR activation is XRCC1 dependent and that PAR and XRCC1 act upstream of resection machinery. These data reveal the involvement of the break repair mechanism in early processing of DNA lesions that is required for the later production of long tract ssDNA (as visualised by RPA foci and non-denatured BrdU staining). This hypothesis is now included in text. We also show that ENDOD1 localisation to chromatin is mediated by PARP1, but does not require PARP3 (Supplementary Figure 2b). This is consistent with the above discussion that the loss of ENDOD1 results in ssDNA that is triggered by inappropriate break repair. We hope these new data provide a sufficient insight into the mechanism for this specific ssDNA production. Importantly, it confirms that the mechanism is distinct from that which occurs upon PARP trapping and progress into S phase (where XRCC1 is not involved).

9- *All the experiments performed in the paper include siRNA and CRISPR mediated KD or KO of the proteins studied. Rescue experiments that restore expression of either p53 and/or ENDOD1 should be included to conclusively prove that the effects seen with synthetic lethality, ssDNA, pRPA, and increased PARylation are not due to off-target effects.*

We now provide rescue experiment data for key experiments, showing the SL and ssDNA production is reliant on concomitant loss of ENDOD1 and TP53. These included in Supplementary Figures 4f, 5d and 6c-e. The specificity of siRNA experiments is also supported by the use of a second siRNA in various experiments.

10- *Mouse xenograft experiments involving characterization of SKOV3 and C33A and full-body targeting of ENDOD1 show that loss of ENDOD1 results in loss of SKOV3/C33A tumor growth. The use of siWDR70 in this experiment as well as characterization of myeloid/lymphoid/bone marrow HSC cells seems mostly tangential to the main message of the story. While this does show that whole-body knockdown of ENDOD1 does not result in gross changes in mouse health, it does not seem to add to showing this growth defect is due to ENDOD1. Use of whole-body knockout of ENDOD1 may have*

significant confounding factors. KD of ENDOD1 throughout the entire animal may significantly alter the tumor microenvironment and therefore the immune response to the implanted tumor cells.

We use the sim*Wdr70* simply as a control as we know that systemic knockdown generates pathology in the animals due to its role in DSB repair. The key point we wanted to examine is whether systemic knockdown of mEndoD1 was similarly toxic. If that were the case, it would be unlikely to be a good SL cancer target as the low toxicity of any SL treatment relies, at least in part, on the drug target. Therefore, we believe these data support that ENDOD1 targeted therapy is worth further investigation. We do agree with referee's concern that systemic knockdown of ENDOD1 may alter the tumour microenvironment, especially given that ENDOD1 was initially found as part of innate immune signalling. In future investigations, and during potential drug development studies, this would be serious consideration.

11- The authors claim that this report is the first to describe evidence of "tumor-specific TP53 mutations could be treated via synthetic lethality". I believe this has already been demonstrated for MAPKAP Kinase-2, in the setting of DNA damaging agents doi:10.1016/j.ccell.2015.09.009).

We thank the referee for drawing our attention to this paper. We agree that previous work has revealed synthetic sensitivity between MAPKAP Kinase-2 and p53 in the context of chemotherapeutic treatments. We now describe our finding as: "These data identify ENDOD1 as a novel cancer-specific target for SL drug discovery."

Other comments:

Fig 1b – PAR intensity graph should also include comparison between RPE1 WT and ENDOD1-/- cells. Zoomed in figure of IF images to better show co-localization of PAR and ENDOD1 would. How many cells were included in the quantification?

Comparison of PAR intensity between RPE1 and ENDOD1-/- is shown Supplementary Figure 2c. Zoomed images for PAR-ENDOD1 colocalisation is shown in Supplementary Figure 2a. Cell number included for quantification was described in methods: In general 150 - 400 cells were quantified per sample in each independent experiment for statistical analysis of imaging assays.

Fig 1e – X-axis label needed

It was labelled as requested.

Fig 1e – Show non-merged images as DAPI merged images make BrdU staining differences hard to visualize

We are sorry we did not identify BrdU staining in Fig 1e.

Fig 2f – How many cells were used in quantification of foci?

It was described in methods.

Fig 3a - How many cells were used in quantification of foci?

It was described in methods.

Fig 4f – X-axis label needed

It was amended.

Fig S1a, right blot – some bands appear to be cropped off on the left side

The marker lane on the left side was restored.

Fig S1b, right blot – some bands appear to be cropped off right side

The gel was extended to the right side.

Fig S1d, cytoplasm fractionation blot

We feel the detection for cytoplasmic fraction of PARPs and histone does not provide further information for understanding the PARP trapping on chromatin.

Fig S2a, b - # events quantified

It was described in methods.

Fig S2c, d – non-merged images so it's easier to see 53BP1 foci and # cells quantified

Separate channels for each staining were provided.

Fig S3a, b – x-axis label needed

It was labelled.

Fig S6a, b – non-merged images and # events quantified

It was amended.

Fig S9c, d, e – MW markers

Markers were labelled.

Reviewer #3

The manuscript by Tang et al identifies a synthetic lethality interaction between TP53 and ENDOD1. The authors document this interaction in both cell lines and mouse models approaches. The findings have the potential for translational significance, since TP53 mutations are frequent in tumors. However, the manuscript brings little mechanistic insights to explain this interaction, and suffers from a lack of rigor making the results unconvincing.

Specific points:

1. The authors make no attempt to investigate the mechanism underlying the ENDOD1-p53 synthetic lethality. It is unclear what the role of p53 is in this context: is this a transcription-dependent or independent function? What are the relevant interactions/targets? How does p53-mediated cell cycle control and apoptosis fit into this? Similarly, how ENDOD1 acts it not addressed. How does it remove PARP1/PAR chains (this is suggested in the model figure). Not even simple experiments (eg if the nuclease activity is involved) are presented. Without a solid understanding of the roles of ENDOD1 and 53 which are underlying the synthetic lethality, the manuscript cannot be published in Nature Communications.

Our manuscript aims to report our discovery and characterisation of SL between ENDOD1 and TP53. A full mechanistic study is beyond the scope of the current work. However, we did characterise the consequence of co-ablation of ENDOD1 and p53 as the generation of ssDNA (which occurs even in G1 arrested cells) and demonstrated the involvement of several factors including PARPs and XRCC1. In the revised manuscript we have added a little more mechanistic data but we still believe that significant future work will be required to understand the full mechanism concerning the p53 and ENDOD1 activities. In the original manuscript we did provide evidence that the role of p53 is independent of its transactivation function and depends on mutations in the DNA binding domain (now Figures 4g, 5e) but, unfortunately, there is little literature about the enzymatic activity of ENDOD1. It is reported as an 'atypical nuclease' and neither its structure or its substrates are known, so it is not yet possible to predict the relevant active site mutant. We further agree that knowing the mechanism of cell death will be interesting, although we believe it is unlikely to involve p53 mediated cell cycle arrest (note: progress into S phase is not required to generate ssDNA upon p53 ablation in ENDOD1^{-/-} cells) or p53-dependent apoptosis.

2. Throughout the manuscript, the authors are employing a single siRNA for both ENDOD1 and TP53. This is a significant weakness, as off target effects for siRNA oligonucleotides are very frequent. (While the authors also employ an ENDOD1-knockout cell line as well as cell lines with p53 mutations, those lines are in general used for a different set of experiments -see comment below). Basically, none of the experiments which are done with siRNA alone can be trusted if only one oligonucleotide is used. This includes all the proliferation experiments. The authors need to perform all these experiments using multiple siRNAs.

We agree with referee's concerns and all key experiments have been validated with a second siRNA.

3. In the same vein: the authors employ a single ENDOD1-knockout clone. As this was obtained by CRISPR, to rule out off-target effects the authors need to employ multiple independent knockout clones, or at the very least complement the knockout clone at hand by re-expressing exogenous ENDOD1.

As described in response to a similar comment from referee 2, we only obtained a single clone of ENDOD1^{-/-} cells. We have addressed the issue by complementing this clone with lentiviral ENDOD1 for several key experiments. Notwithstanding this, we have also used two coisogenic cell systems (SKOV3 and HCT116) to validate the ENDOD1 - TP53 SL and performed siRNA experiments (with multiple siRNAs) in a wide variety of cell lines.

4. The authors employ ENDOD1 siRNA and knockout approaches throughout the manuscript in a seemingly random manner. For example, the comet and drug sensitivity assays are done with the knockout clone, and the proliferation experiments are done with siRNA. What is the rationale for this approach? All important experiments should be performed with both knockdown and knockout.

We present ENDOD1-TP53 SL with siENDOD1 vs ENDOD1^{-/-} and siTP53 (Figure 3c-d and Supplementary Figure 4c), as well as siENDOD1 with TP53 mutants or null cancer lines (Figure 3a and Supplementary Figure 4a, 4b). We also provide SL data for coisogenic HCT116 harbouring wildtype or mutant TP53 (Supplementary Figure 4h). We employed these combined treatments with different cell lines and reagents for the purpose of cross supporting the SL and signal generation.

5. Another major weakness is the use of “proliferation” experiments to measure synthetic lethality. The authors employed hemocytometer counting for these experiments, but this is an inaccurate method. Since the differences reported are minor (eg Fig 1d, 2a), these results are not convincing. The authors need to rigorously measure survival/proliferation using a variety of different assays, including clonogenic assays and measurement of cell proliferation using MTT, CellTiterGlo or similar reagents.

While we believe Haemocytometer counting provides a straightforward, direct, accurate and rigorous assessment of cell proliferation, we do agree that the data will be strengthened by using multiple methods. We have therefore applied the CCK8 assay for many of the proliferation experiments in the revised manuscript.

6. On page 3 when discussing fig 2, the authors state that “concomitant loss of p53 and ENDOD1 results in de-novo generation of ssDNA”. What does “de novo” mean in this context? Do the authors imply that this occurs during DNA replication? This would in fact be an important point since it may allow some mechanistic insights, so the authors should address it by labeling cells with BrdU and investigating ssDNA formation in BrdU-labeled cells.

By de-novo we simply meant that it was not present before treatment and arose anew. We hope our expanded manuscript now makes clear that the ssDNA can arise in G1 phase in cells that are not able to enter S phase – i.e. cells unable to incorporate BrdU, as shown in Supplementary Figure 4g (this experiment was performed in parallel with the experiment shown in Supplementary Figure 5c, differing in BrdU detection: denatured, non-denatured). Further mechanistic insight is given in the revised manuscript by studying the requirement for various additional factors such that we now show (Figure 6) that ssDNA production and PAR activation is XRCC1 dependent and that PAR and XRCC1 act upstream of resection machinery. These data reveal the involvement of the SSBR repair mechanism in the early processing of DNA lesions that is required for the production of long tract ssDNA (i.e. detectable by RPA and PAR staining). We hope our new data provide a sufficient insight into the mechanism for ssDNA production. Importantly, it confirms that the mechanism is distinct from that which occurs upon PARP trapping and progress into S phase (where XRCC1 is not involved).

7. Does loss of PARP1 restore DNA resistance to S1 nuclease in the P53-ENDOD1 codepleted cells? This would be important to assess, in order to establish the ssDNA accumulation as the cause of the synthetic lethality.

We thank the reviewer for this insightful question and performed the relevant experiments. siPARPs and PARPi partially rescued S1 sensitivity in ENDOD1-TP53 double mutants (Supplementary Figure 6h). We interpret this to indicate that, besides extensive ssDNA, i.e. as seen by RPA staining, other forms of damage (i.e. nicks/gaps) are initially present in ENDOD1-TP53 double mutant cells, but that these are less toxic. However, they are prone to processing in a PARP- and XRCC1-dependent manner. It is notable that this may correspond to the only partial rescue of viability seen upon PARPi treatment (Figure 5g).

8. What is the relevance of the “truncated nuclear isoform” presented in figure 1? No studies are presented to identify the exact nature of the band on the blot, and if it is functionally relevant. Thus it makes no sense to introduce and discuss it. The authors write in the third line of the Discussion section that they “identify an additional role for a nuclear isoform of ENDOD1 in regulating PARP” but there is absolutely no evidence of this presented. The immunofluorescence experiment in Fig 1b is performed with an antibody recognizing the full length ENDOD1, so it cannot be implied that the co-

localization described is with the truncation.

We apologise for the poor presentation. In the new manuscript we focus on the phenotypic effects of loss of ENDOD1. In the initial submission we simply wanted to emphasise that there are multiple isoforms and that some ENDOD1 is found in the nucleus, which is consistent with a role in DNA repair. We have now deleted the fractionation experiments as their preliminary nature does not add to the story.

9. Fig 2b does not seem to present the quantification of the colocalization between PAR and ENDOD1, but rather the quantifications of each of them separately. Just because they follow the same trend, it cannot be concluded that they co-localize.

We agree that the data did not show colocalization, but that they overlap. This is now shown in Supplementary Figure 2a.

10. The statistical analyses for the comet experiments (eg Fig 1c, 2c) are missing. How many independent experiments were performed? How were the data analyzed to ensure that the effects observed are statistically significant?

This was done with 3 biological repeats and the statistical tools are mentioned in the figure legend.

11. In order to address the clinical relevance of these findings, the authors should explore TCGA and other similar databases for evidence of this synthetic lethality. For example, one would anticipate that mutations in TP53 and ENDOD1 are mutually exclusive in tumors. Is this the case?

We have looked at this, but the results are hard to interpret as it is impossible to define the pathological mutations of ENDOD1. Correlation based on random mutation does not provide evidence for mutual exclusivity.

12. The model in Fig 3g is not described anywhere, not in the text nor in the figure legend. It is unclear how the model is supported by the data presented, or if this is a purely fictional model.

The model is now referred to in the text.

13. The manuscript is written in a very concise manner, which is not beneficial to the paper. It is unclear why each of the experiments were performed, what was the rationale, and what was gained from that particular experiment. The reader is left to guess the take home message

We apologise for the poor presentation and have significantly extended the experimental descriptions and logic for the various experiments.

REVIEWERS' COMMENTS

Reviewer #1 (Remarks to the Author):

The authors have responded to the reviewers reports in a comprehensive way and the manuscript is greatly improved. The data support a novel function for p53 in DNA repair and an SL interaction with ENDOD1. This is a very important observation not only in the identification of a ENDOD1 as a novel drug target but also in our basic understanding of how p53 functions as a tumor suppressor. My particular issues as a reviewer have been answered .

Reviewer #3 (Remarks to the Author):

The revised manuscript addresses a number of my main concerns, in particular those dealing with technical issues (additional controls, siRNAs, correction of the knockout cell lines etc). I find that the manuscript is now technically sound and rigorous.

While some additional insights are shown regarding the mechanism of the observed synthetic lethality, this previous major comment I had is still not satisfactorily addressed. Nevertheless, the potential clinical significance of the observed genetic interaction goes some way to alleviate my concerns regarding this lack of mechanistic understanding.

Moreover, recently published studies have uncovered an unexpected role of ssDNA in modulating cellular viability and drug sensitivity in BRCA-deficient cells (PMID: 34508659, 34624216, 34358459, 33184108, 34555355 and others). These findings may be relevant to the genetic interaction observed here. At the very least, the authors should cite these papers, and discuss their results in the context of this recently-published work on ssDNA gaps.

14-04-2022

Response to referee NCOMMS-21-01761A

We thank the reviewers for again reading and commenting on our manuscript. We are pleased that the reviewers feel the work is now acceptable for publication in Nature communications. Referee 3 stated:

'Moreover, recently published studies have uncovered an unexpected role of ssDNA in modulating cellular viability and drug sensitivity in BRCA-deficient cells (PMID: 34508659, 34624216, 34358459, 33184108, 34555355 and others). These findings may be relevant to the genetic interaction observed here. At the very least, the authors should cite these papers, and discuss their results in the context of this recently-published work on ssDNA gaps.'

While we are aware of this literature, after careful consideration we have not included further discussion of the recent literature on BRCA1 and single strand lesions as we did not feel this was helpful to the reader and, since it appears to be a BRCA-specific issue and the majority of our discussion centres around the p53 synthetic lethality. Indeed, we did not find a place where such a discussion could realistically fit.

Prof A M Carr FMedSci (Director)
Prof K W Caldecott FMedSci (Deputy Director)
Dr J Baxter
Dr A Bianchi
Dr K-L Chan
Prof A Doherty
Dr H Hoehegger
Prof P A Jeggo FMedSci
Prof A R Lehmann FRS FMedSci

Dr J M Murray
Dr M Neale
Prof M O'Driscoll
Dr A W Oliver
Prof L H Pearl FRS FMedSci
Prof Uli Rass
Prof Evi Soutoglou

Genome Damage and Stability Centre
University of Sussex, Falmer,
Brighton BN1 9RQ, United Kingdom

T +44 (0)1273 678123
F +44 (0)1273 678121
E gdsc@sussex.ac.uk

www.sussex.ac.uk/gdsc